# Repeated dosing improves oncolytic rhabdovirus therapy in mice via interactions with intravascular monocytes

Victor Naumenko [1,2,3,4✉], Jahanara Rajwani[1,2,5], Madison Turk [3,6], Chunfen Zhang[1,2], Mandy Tse[3,6], Rachelle P. Davis[3,6], Daesun Kim[1,2], Andrea Rakic[1,2], Himika Dastidar[1,2], Shinia Van[1,2,3], Laura K. Mah[2,3,6], Esha K. Kaul[2,3,6], Vladimir P. Chekhonin[4,7], Douglas J. Mahoney [1,2,5,6,8✉] & Craig N. Jenne [3,6,8✉]

There is debate in the field of oncolytic virus (OV) therapy, whether a single viral dose, or multiple administrations, is better for tumor control. Using intravital microscopy, we describe the fate of vesicular stomatitis virus (VSV) delivered systemically as a first or a second dose. Following primary administration, VSV binds to the endothelium, initiates tumor infection and activates a proinflammatory response. This initial OV dose induces neutrophil migration into the tumor and limits viral replication. OV administered as a second dose fails to infect the tumor and is captured by intravascular monocytes. Despite a lack of direct infection, this second viral dose, in a monocyte-dependent fashion, enhances and sustains infection by the first viral dose, promotes CD8 T cell recruitment, delays tumor growth and improves survival in multi-dosing OV therapy. Thus, repeated VSV dosing engages monocytes to post-condition the tumor microenvironment for improved infection and anticancer T cell responses. Understanding the complex interactions between the subsequent viral doses is crucial for improving the efficiency of OV therapy and virus-based vaccines.

[1] Alberta Children's Hospital Research Institute, Calgary, AB T2N 4N1, Canada. [2] Arnie Charbonneau Cancer Institute, Calgary, AB T2N 4N1, Canada. [3] Snyder Institute for Chronic Disease, Calgary, AB T2N 4N1, Canada. [4] V. Serbsky National Medical Research Center for Psychiatry and Narcology, Moscow 119034, Russia. [5] Department of Biochemistry and Molecular Biology, Faculty of Medicine, University of Calgary, Calgary, AB T2N 4N1, Canada. [6] Department of Microbiology, Immunology and Infectious Disease, Faculty of Medicine, University of Calgary, Calgary, AB T2N 4N1, Canada. [7] Department of Medical Nanobiotechnology, N.I Pirogov Russian National Research Medical University, Moscow 117997, Russia. [8] These authors contributed equally: Douglas J. Mahoney, Craig N. Jenne. ✉email: naumenko.vict@gmail.com; djmahone@ucalgary.ca; cnjenne@ucalgary.ca

Oncolytic virus therapy (OVT) is an approach to treating cancer that employs live viruses to eliminate tumor cells. Although initially believed to mediate much of its therapeutic effect through the direct infection and lysis of tumor cells, our understanding of OVT has, for the most part, shifted to the immunomodulatory functions of the viral infection[1,2]. Currently approved OVT (H101 and Imlygic), and many of the therapies presently under clinical evaluation, rely on intratumoral injection of the virus[3,4]; however, intravenous administration could enable OV access to broader range of cancers including metastatic lesions, though delivering sufficient amounts of infectious virus to ensure adequate infection remains challenging. Due to a high viral uptake rate by reticuloendothelial system and low extravasation efficiency in tumor beds, only around 0.001–0.01% of a systemically injected dose accumulates in tumor lesions[5,6]. Even if OV reaches a given tumor site, viral distribution in the tumor mass is nonuniform and remains mostly limited to regions located with 50 μm of blood vessels[7]. Critically, immune cells and molecules (antibodies) have the capacity to inactivate viruses within the bloodstream and clear foci of infection in tumor, even in virus-naïve hosts.

Low efficiency of OV delivery to tumor cells after systemic administration can be partly compensated through the injection of a higher viral dose; however, safety concerns limit effectiveness of such an approach. Instead, to deliver more virus to tumors without eliciting undesired side effects, multiple dosing protocols are used. The time between treatments depends on OVT tolerability, and cycle duration is usually determined by the development of an adaptive immune response (approximately 14 days in humans). In clinical trials, Reolysin and Parvoryx are injected intravenously (i.v.) for five consecutive days[8–10]. In other protocols Reolysin is administered on day 1, 8, 15, and 22 (NCT03015922), or as 2 consequent doses on days 1–2, 8–9[11], and 15–16 ([12], NCT03605719). Newcastle disease virus treatment schedule consists of 4–6 doses injected i.v. within 2 weeks[13,14]; coxsackie virus is administered on day 1, 3, 5, and 8 (NCT02824965) and 3–4 shots of vaccinia virus are given once a week (NCT03294486, NCT03294083). Interestingly, in multiple clinical trials, vesicular stomatitis virus (VSV)-IFN-β is used as a single dose therapy (NCT03647163, NCT03120624, NCT02923466, NCT03017820, and NCT04291105), while another oncolytic rhabdovirus – Maraba – is administered as 2 or 4 i.v. injections with 3-4 days between subsequent doses (NCT02285816; NCT02879760). This diversity in treatment regimens demonstrates the need for a better understanding of viral dynamics following systemic administration of a multidose therapy.

Although in practice, a majority of OV are given as repeat treatments, the mechanisms responsible for the improved survival in multidosing protocols are poorly understood[15]. When initially designed, these multidosing approaches based on the possibility that each consequent treatment increases the chances for OV to infect and kill more tumor cells; however, it is also likely that administration of the first dose of virus will influence and modulate the response to the second viral administration. It is well established that systemic injection of live virus activates inflammatory and immune cells[16–18], elicits a proinflammatory cytokine response[19–21] and may lead to vascular collapse in tumors[6]. This initial response means that subsequent doses of virus face a fundamentally different tumor microenvironment that could either restrict or enhance the antitumor activity of OV. Elucidating the potential interplay between repeat OV doses and their interaction with the host immune system and tumor cells is crucial for improving OVT efficiency.

We have developed techniques for fluorescent labeling of oncolytic rhabdoviruses and for tracking their microdistribution, infection and interactions with host cells in tumor and lymphoid organs in vivo[22]. Here, using intravital microscopy (IVM), we describe the behavior of VSV$^{\Delta M51}$, an attenuated viral strain that is highly susceptible to the host interferon response, delivered systemically as a first or a second dose. Multiple features of this virus make it an attractive agent for cancer therapy: oncolytic activity against a broad range of cancers; safety for normal cells; cytoplasmic replication without risk of host cell transformation; a lack of pre-existing immunity; a small and easy to manipulate genome, and scalability of production[23]. Adding to these advantageous features, our results indicate subsequent doses of VSV$^{\Delta M51}$ – despite limited direct oncolytic activity – potentiate the infection and antitumor response of the initial viral dose via complex interactions with immune cells of tumor microenvironment.

## Results

**Secondary dose of OV does not infect the tumor but enhances infection of the initial OV treatment.** The dynamics of both the initial and second VSV$^{\Delta M51}$ dose were studied in immuno-competent mice implanted with a subcutaneous (s.c.) tumor of CT-26$^{LacZ}$ colorectal carcinoma. This model is highly sensitive to VSV$^{\Delta M51}$ infection[24] making CT-26$^{LacZ}$ tumors a reproducible model for studying infection dynamics. We have shown that a single i.v. injection of $5 \times 10^8$ plaque-forming units (PFU) VSV$^{\Delta M51}$ cures 100% of animals bearing CT-26$^{LacZ}$ tumors[22]. To better capture potential differences between a single and repeated treatment model, we decreased the viral dose to $1 \times 10^6$ PFU. Under these conditions, two i.v. injections, with a second dose 48 h after the initial treatment, delayed tumor growth and improved survival more efficiently than a single dose (Fig. 1a, b). These results were validated in animals bearing a s.c. M3-9-M rhabdomyosarcoma treated with VSV$^{\Delta M51}$ ($5 \times 10^8$ PFU) with or without a second treatment dose (Supplementary Fig. 1a, b). To track the effect of multiple doses of OVT on viral infection of the tumor, we quantified VSV-delivered firefly luciferase (VSV$^{\Delta M51}$-FLUC) by whole-body bioluminescence imaging. Infection in the tumor after a single dose peaked at 48 h post-infection (hpi) and began to wane by 72 hpi. A second dose of OVT 48 h after the initial dose resulted in more pronounced and sustained viral replication in the tumor (Fig. 1c). Robust tumor infection was also observed by IVM following administration of two doses of OVT containing a GFP reporter gene (Fig. 1d). As these approaches only approximate infection by measuring delivery of a luciferase or a GFP gene, we determined the TCID$_{50}$ of tumor homogenates and confirmed two doses of OVT resulted in increased infection (Fig. 1e).

To assess the specific contribution of each dose of OVT to tumor infection, we utilized an approach whereby only one dose of OVT (either first or second) contained a luciferase gene. With this strategy we could specifically measure what portion of the infection can be attributed to each dose. Administration of VSV$^{\Delta M51}$-FLUC followed by VSV$^{\Delta M51}$-NR resulted in robust bioluminescence. In contrast, administration of VSV$^{\Delta M51}$-NR followed by VSV$^{\Delta M51}$-FLUC resulted in minimal bioluminescence (Fig. 1f, g). These results suggested that nearly all observed tumor infection was mediated by the virus delivered in the first dose. Similarly, localized foci of infection could be detected in M3-9-M tumors after initial VSV$^{\Delta M51}$ dose ($5 \times 10^8$ PFU), but not following the second dose (Supplementary Figure 1c). To ensure this observation was not due to an artifact associated with the luciferase reporter system, we repeated the experiment using VSV$^{\Delta M51}$-delivered GFP and determined viral infection using IVM, confirming that the second dose of OVT failed to infect the tumor in any considerable fashion (Fig. 1h). Importantly,

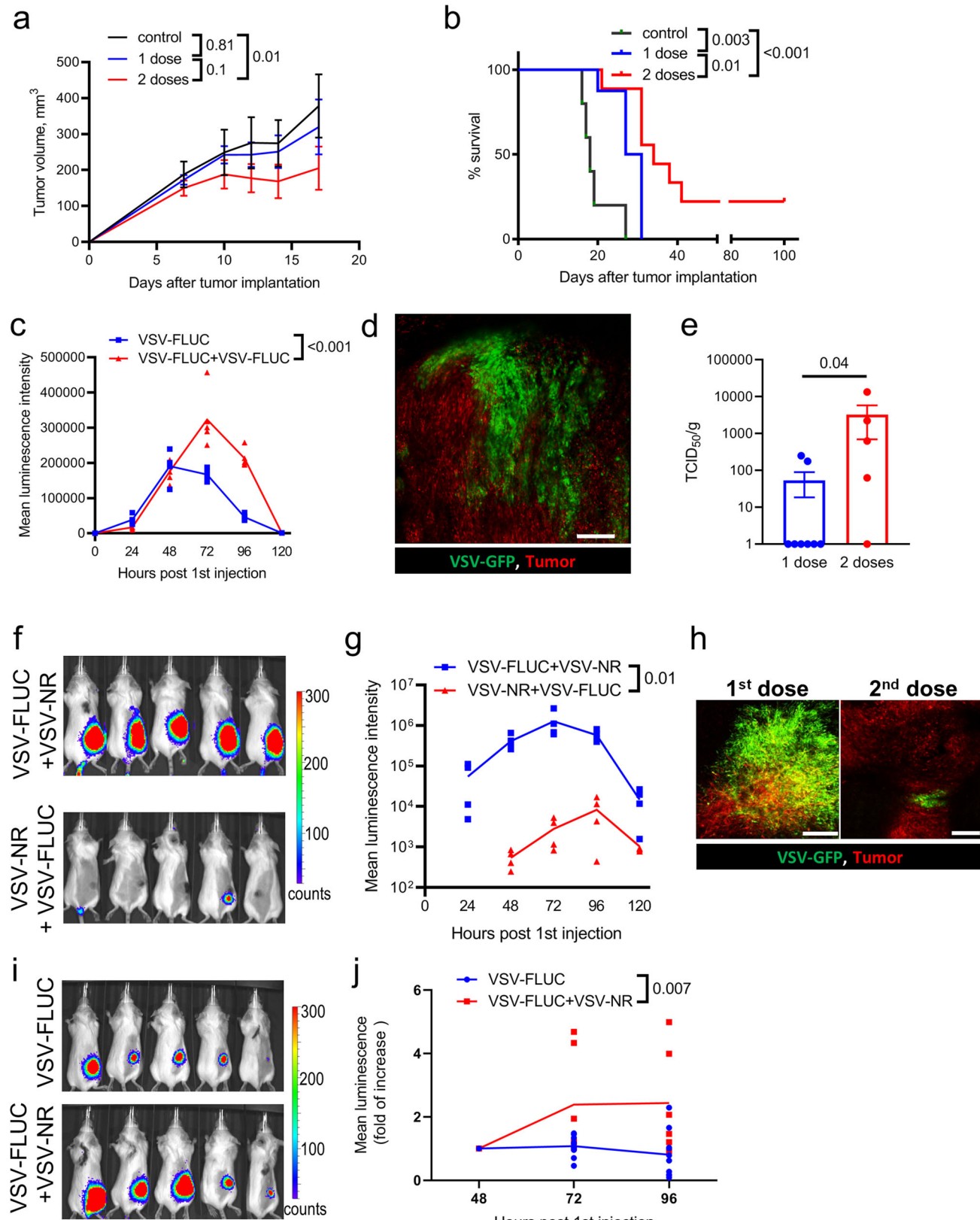

although the second dose of OVT did not infect the tumor, its administration improved and sustained infection of the tumor by the first dose (Fig. 1i, j). These results were reproduced following a similar treatment schedule using Maraba MG1[25,26], another rhabdovirus that is currently in clinical trials (Supplementary Figure 1d, e). Extending this observation of immune modulation

by a second dose of virus, we also determined if the third dose could further enhance the infection and found there was no difference in tumor bioluminescence between mice treated with two or three doses of VSV (Supplementary Fig. 1f).

$VSV^{\Delta M51}$ is also known to transiently replicate in lymphoid organs[22,27,28]. Interestingly, examination of spleen and lymph

**Fig. 1 OV delivered as a second dose does not infect the tumor yet enhances the infection of the initial OV treatment. a** Tumor measurements for CT26$^{LacZ}$-bearing untreated animals ($n = 6$) and mice injected with either one ($n = 6$) or two doses ($n = 8$) of VSV ($10^6$ PFU, 48 h between treatments). Results are shown as mean ± SEM; two-way ANOVA followed by Sidak's multiple comparisons test. **b** Kaplan–Meier survival plots for CT26$^{LacZ}$-bearing untreated animals ($n = 5$) and mice injected with either one ($n = 8$) or two doses ($n = 9$) of VSV; log-rank test. **c** Luminescence intensity for CT26$^{LacZ}$ tumors after single or repeated VSV-FLUC dosing ($10^6$ PFU, 48 h between doses, $n = 5$). Results are shown as mean with individual values; two-way ANOVA. **d**. Representative IVM image of infection (green) in CT26$^{LacZ}$ tumors (red) 24 h after a second VSV-GFP dose ($10^6$ PFU, 48 h between treatments). Scale bar, 200 μm. **e** Virus titers in CT26$^{LacZ}$ tumors at 72 h post single dose treatment ($n = 8$) or 24 h post second VSV-GFP dose ($10^6$ PFU; $n = 5$) measured as TCID$_{50}$/g tissue. Results are shown as mean ± SEM; Mann–Whitney test. **f** Representative bioluminescence images for the groups shown in (**g**) at 24 h post first or second VSV-FLUC dose. **g** Luminescence intensity for CT26$^{LacZ}$ tumors after first or second VSV-FLUC i.v. injection ($10^6$ PFU, 48 h between doses, $n = 4$). To discriminate between first and second dose infection, VSV-NR was used as the second or first treatment, respectively. Results are shown as mean with individual values; two-way ANOVA. **h** Representative IVM images of infection (green) in CT26$^{LacZ}$ tumors (red) 24 h after first or second VSV-GFP dose ($10^6$ PFU, 48 h between treatments). To evaluate the impact of the second dose in overall infection, VSV-NR was used as the first treatment in the 2-dose schedule. Scale bar, 200 μm. **i** Representative bioluminescence images for the groups shown in **j** at 96/48 h post first/second OV treatment. **j** Luminescence intensity for CT26$^{LacZ}$ tumors after VSV-FLUC treatment ($10^6$ PFU) with or without VSV-NR injection ($10^6$ PFU) coming 48 h later ($n = 6$). Luminescence intensities for individual tumors are normalized to the mean luminescence intensity in each group at the time of second dosing. Results are shown as mean with individual values; two-way ANOVA.

nodes revealed that the second OVT dose failed to infect these lymphoid tissues altogether with no increased or sustained infection observed (Supplementary Fig. 1g–i).

Collectively, these results suggest the second dose of OVT fails to infect, but instead supports the ongoing tumor infection mediated by the initial OVT dose.

**OV administered as a second dose interacts with intravascular leukocytes recruited to the infected tumor**. To understand how a second dose of OVT acts to increase and prolong tumor infection by a prior dose, we utilized IVM and fluorescently labeled virus to track the fate of i.v. administered VSV[22]. Whereas the first dose of virus primarily bound to vascular endothelium, intravascular neutrophils, and tumor cells (Fig. 2a–d, i and Supplementary Movie 1), virus in the second dose was rapidly captured by a different intravascular leukocyte population (Fig. 2e–i). Inclusion of additional cell markers identified these cells as Ly6g$^-$, CD11b$^+$, Ly6c$^+$, F4/80$^+$, CD169$^+$ (Fig. 2e–h and Supplementary Movie 2). These cells remained intravascular and were observed to be mobile, crawling along the tumor endothelium, suggesting they represented a population of monocytes.

Flow cytometric (FC) analysis allowed for more detailed characterization of the cells binding the VSV virions within the blood (Supplementary Fig. 2a) and in the spleen (Supplementary Fig. 2b). Although some virus binding by blood leukocytes could be observed following the first dose of OVT, viral binding by CD11b$^+$Ly6C$^{hi}$ cells was markedly enhanced following administration of the second dose. Likewise, binding of the initial dose of OVT by macrophage was observed in the spleen; however, a second dose of OVT 48 h after the initial treatment resulted in a pronounced shift in viral binding with reduced capture by CD11b$^+$CD169$^+$ macrophages and enhanced viral binding by CD11b$^+$Ly6C$^+$ splenocytes (Supplementary Fig. 2b).

In addition to a shift in the phenotype of cells binding virus between the first and second dose of OVT, the initial tumor infection also induced major changes in the relative composition of intravascular leukocyte populations within the tumor. To characterize intravascular cells within the tumor by FC, we utilized a technique where anti-CD45 antibodies were injected i.v. 10 min before collecting tumor tissue. This technique has been demonstrated to differentially label intravascular cells, making it possible to identify the location of cells within the tumor microenvironment, intravascular vs. interstitial cells[29]. Using this approach, we identified an accumulation of intravascular macrophages 48 h following the initial OVT dose (Fig. 2j). These alterations in tumor intravascular populations were also reflected

by shifts in the frequency of circulating leukocytes, with an increased frequency of multiple CD11b$^+$ cell populations within the peripheral blood and spleen 48 h following OVT administration (Fig. 2k and Supplementary Fig. 2c, d). Observations within the vascular compartment were paralleled within the tumor interstitium, with an increased frequency of CD11b$^+$ monocyte/macrophage populations observed 48 h following administration of VSV$^{\Delta M51}$ (Fig. 2l). Analysis of intratumor lymphocyte populations did not reveal significant shifts in relative cell frequencies 48 h following the initial dose of $10^6$ PFU VSV$^{\Delta M51}$ (Supplementary Fig. 2e). We hypothesized that the second dose further activates monocytes, however there was no change in number of these cells in blood or tumor 24 h following virus rechallenge (Supplementary Fig. 2f).

**Interactions between the second dose of OV and intravascular monocytes promote sustained infection by the initial OV dose**. Given intravascular monocytes were observed to bind the second dose of OVT, we explored the possibility that these cells acted as a sink, absorbing virus and preventing it from accessing the tumor, thus explaining why we failed to detect tumor infection by the second dose of VSV$^{\Delta M51}$. To test this hypothesis, we used either clodronate liposomes (CLL) or anti-CCR2 antibodies[30] to deplete monocytes/macrophage 24 h after the initial dose of OVT but prior to injection of the second dose. While CLL depleted both monocytes and F4/80$^+$ macrophages, anti-CCR2 selectively eliminated monocytes in blood, spleen, and tumor (Supplementary Fig. 3a–e). To assess the impact of monocytes on the infectivity of the second OV dose, CLL was administered 24 h after VSV$^{\Delta M51}$-NR and 24 h later mice were administered VSV$^{\Delta M51}$-FLUC. Monocyte depletion did not improve the infection of the second dose (Fig. 3a, b). When VSV$^{\Delta M51}$-FLUC was used both as a first and a second treatment, depletion of monocytes by CLL (Fig. 3c) or by administration of anti-CCR2 (Fig. 3d) resulted in less overall infection of the tumor, indicating the enhanced and sustained infectivity observed in response to the second OV dose was inhibited in monocyte-depleted animals. Moreover, in the absence of monocytes, we observed reduced tumor clearance and overall survival (Fig. 3e, f). Importantly, monocyte depletion did not impact tumor infection following the first dose of VSV$^{\Delta M51}$ (Supplementary Fig. 3f, g). These results indicate that i) monocytes do not affect infectivity of the second dose and, ii) do not enhance tumor infection in single dose treatment schedule; however, these cells are needed to facilitate the overall biological effect of the multiple-dose treatment regime of OVT.

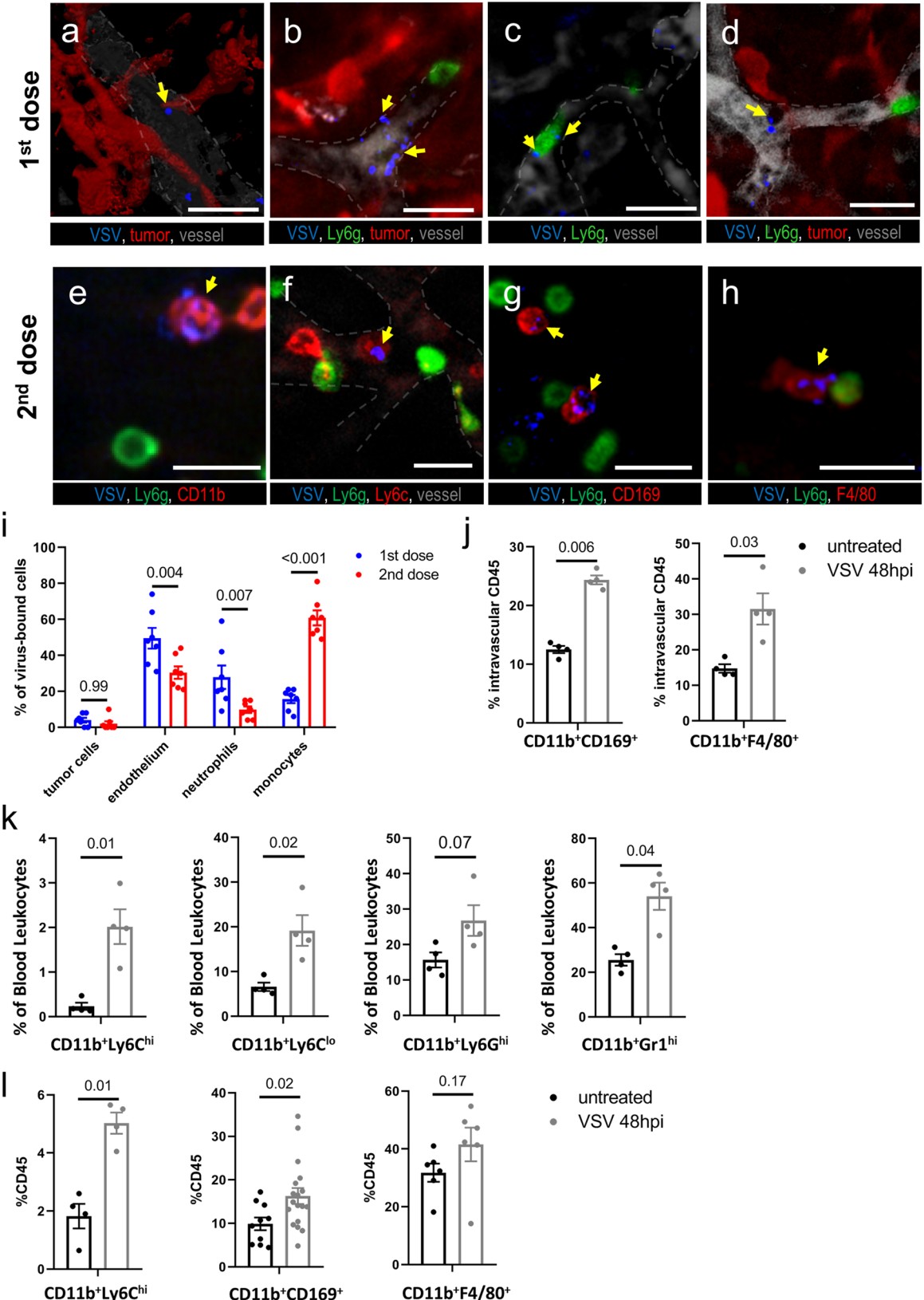

**Second dose of OVT limits antiviral activity of neutrophils through a monocyte-dependent mechanism**. To understand how repeat VSV$^{\Delta M51}$ dosing might affect viral productivity in the tumor, we quantified cytokines and chemokines in tumor and blood. The second OVT dose reduced levels of the chemokine MIP-2 both locally (tumor interstitial fluid) and systemically (serum) (Fig. 4a, b). Given MIP-2 is a chemoattractant and activator of neutrophils[31], we next examined neutrophil recruitment to, and behavior within, the tumor. IVM analysis identified neutrophil accumulation in infected areas of the tumor within hours of OVT administration (Fig. 4c and Supplementary Movie 3). Interestingly, the number of neutrophils within the

**Fig. 2 Intravascular leukocytes recruited to OV-infected tumors interact with OV administered during a second dose. a–h** Intravital imaging of CT26$^{LacZ}$ tumor microenvironment after initial (**a–d**) or second dose (**e–h**) of VSV-AF647. Virus (blue, arrows) delivered as a first dose interacts with tumor cells (**a**), endothelium (**b**), neutrophils (**c**), and other leukocytes (**d**). Virus delivered as a second dose is mainly captured by monocytes defined as CD11b$^+$Ly6G$^-$ cells (**e**), expressing Ly6C (**f**), CD169 (**g**), and F4/80 (**h**). Vessels counterstained with FITC-BSA (gray) and delineated by white dashed lines. Scale bar, 25 μm. **i** Quantification of VSV microdistribution in tumor microenvironment after single or repeated VSV administration. Results are shown as percentage of total VSV-bound cells and plotted as mean ± SEM ($n = 7$; two-way ANOVA followed by Sidak's multiple comparisons test). **j** FC analysis of intravascular leukocytes in tumor samples at the time of first or second VSV treatment. Intravascular fraction of leukocytes is identified by anti-CD45 injected into the tail vein 10 minutes before tissue collection. Results are shown as percentage of intravascular cells and plotted as mean ± SEM ($n = 4$; unpaired t-test). **k** FC analysis of blood samples collected from untreated mice or 48 h post VSV i.v. injection (10$^6$ PFU). Results are shown as percentage of CD45 + cells and plotted as mean ± SEM ($n = 4$; unpaired t-test). **l** FC analysis of select leukocyte populations in CT26$^{LacZ}$ tumor at the time of first or second VSV treatment (see also Supplementary Fig. 2e). Results are shown as percentage of CD45 + cells and plotted as mean ± SEM (unpaired $t$-test).

tumor returned to levels comparable to that of untreated controls 24 h following a second dose of OV. This reduction in intratumor neutrophils following the second viral dose was dependent on monocytes, as treatment of mice with anti-CCR2 between the first and second doses of OV abrogated this reduction in intratumor neutrophils (Fig. 4d). This loss of neutrophils was a tumor-specific phenomenon as circulating neutrophil populations were not significantly impacted following OVT (Supplementary Fig. 4a).

IVM analysis of cell behavior within the tumor interstitium revealed a reduction in the percentage of neutrophils crawling through the tissue, and an increase in the percentage of stationary cells (Fig. 4e). This shift in cell behavior was dependent on the presence of monocytes, as anti-CCR2 treatment administered between the first and second OV doses prevented this behavioral shift. Moreover, following the second viral dose, we observed an increase in the number of neutrophils forming stable interactions with intravascular monocytes, suggesting a direct reprogramming of neutrophil activation within the tumor vasculature (Fig. 4f, g and Supplementary Movie 4).

As inflammatory cells participate in the generation of an antiviral state within the tumor, we proposed that the ability of the second dose of VSV$^{\Delta M51}$ to support and enhance the infection mediated by the initial OVT treatment might be due to modulation of neutrophil-driven inflammation and antiviral immunity. To test this theory, we examined viral infection and tumor progression in neutrophil depleted mice. Treatment of animals with anti-Ly6g antibody resulted in greater than a 99% reduction in both circulating and intratumoral neutrophils (Supplementary Fig. 4b, c). Depletion of neutrophils 24 h following a single dose of OV enhanced viral infection of the tumor (Fig. 4h). Despite this increased tumor infection, neutrophil depletion did not improve tumor clearance or survival following a single OVT dose (Fig. 4i, j). Administration of a second OVT dose after neutrophil depletion further enhanced tumor infection (Supplementary Fig. 4d–f) and resulted in a small but significant reduction in tumor volume and increase in survival (Fig. 4k, l). These results suggest the recruitment of neutrophils to the infected tumor limits viral infection but does not itself impair tumor clearance.

**Monocyte-dependent CD8$^+$ cell accumulation in tumor following multiple doses of OVT.** A multiple dose regimen of OVT results in attenuation of tumor growth and improved animal survival (Fig. 1a, b). Previous studies have demonstrated a critical role for CD8$^+$ T cells in the antitumor response generated by OVT[19,32,33]. To determine if enhancement of infection by the second dose supports increased survival through direct antitumor activity or through modulation of the CD8$^+$ T cell response, we used an antibody-based depletion strategy to reduced CD8$^+$ cell counts in the lymph node, spleen, and tumor by at least 98% (Supplementary Fig. 5a–c). In these CD8$^+$ T cell-depleted

animals, the protective effect of OVT is completely abrogated (Fig. 5a, b). Further evidence of the role of CD8$^+$ T cells in tumor clearance is provided by studies where tumors are re-introduced into animals that have previously cleared tumors (Supplementary Fig. 5d). In these experiments, the implanted tumor is quickly eliminated, demonstrating that the initial tumor clearance generated immune memory that facilitated a rapid and robust recall response.

In addition to the antitumor response, CD8$^+$ T cells also contribute to the antiviral response. Depletion of CD8 + T cells results in a more robust and sustained OV infection of the tumor (Fig. 5c, d). Interestingly, pretreatment with anti-CD8 antibodies led to the drop in the levels of proinflammatory cytokines and chemokines in TIFs (Supplementary Fig. 5e) that may potentially explain the enhancement of tumor infection. Of note, although CD8$^+$ T cells limit viral infection, this same infection is responsible for enhanced CD8$^+$ T cell accumulation within the tumor (Fig. 5e). Following the second dose of OV, a pronounced accumulation of CD8$^+$ T cells were observed within the tumor interstitium (Fig. 5f). Despite this accumulation of CD8 + T cells within the tumor, we did not observe any significant difference in cellular behavior (motile, stationary, etc.) of CD8$^+$ T cells after administering the repeated VSV dose (Supplementary Figure 5f). Depletion of monocytes after the first dose of VSV attenuated the observed CD8$^+$ T cell accumulation following the second viral dose (Fig. 5e–g). Importantly, this impact on CD8$^+$ T cell numbers appears to be restricted to the tumor, as depletion of monocytes by anti-CCR2 treatment did not impact blood CD8$^+$ T cell frequencies following either first or second dose of OVT (Supplementary Fig. 5g).

## Discussion

It has been reported that multi-dose OVT protocols result in improved tumor clearance, a response that is usually attributed to enhanced tumor infection[24,34,35]. This multiple dose regimen is classically believed to deliver more virus to the tumor, increasing the quantity of infection while reducing the risk of an acute response (cytokine storm) to a large bolus of virus given as a single dose – essentially yielding an additive infectious load. In the current study, tumor infection and antitumor responses were observed to be higher upon systemic administration of two VSV$^{\Delta M51}$ doses as compared to a single dose treatment regime in CT-26$^{LacZ}$ and M3-9-M tumor models. Evaluation of the contribution of each dose of OVT to overall tumor infection demonstrated that the ability of the second viral dose to infect the tumor is negligible, suggesting antitumor activity is mediated by some other mechanism(s).

The current study demonstrates that the antitumor activity of OV delivered as repeat doses fundamentally differs from the well-described mechanisms of single-dose OV treatment. The first OV dose infects cancer cells (Fig. 6), activating monocytes, driving neutrophil recruitment to tumors, and eliciting systemic and local

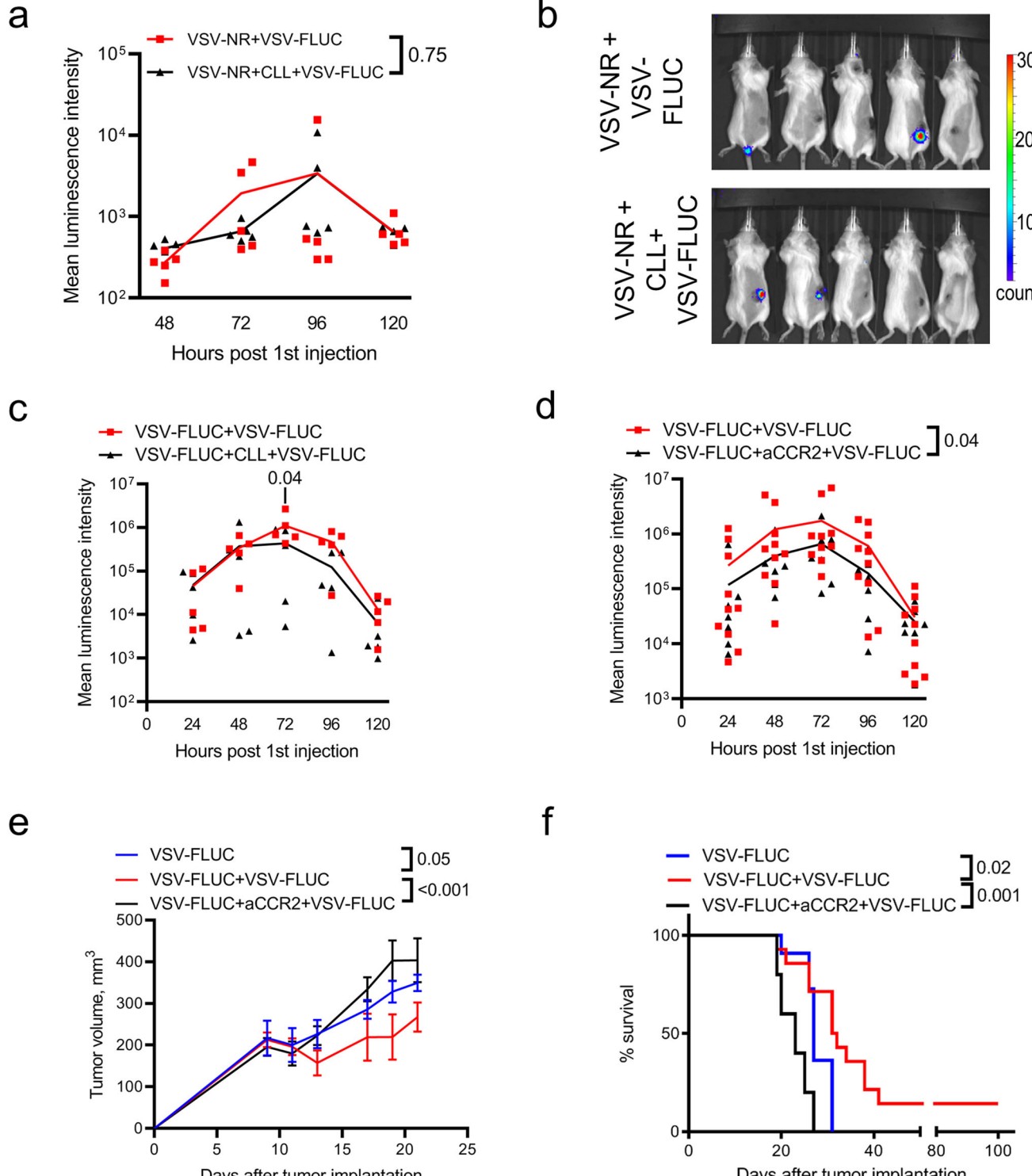

**Fig. 3 Interactions between OV and inflammatory monocytes promote better infection of the initial OV dose. a** Luminescence intensity for CT26[LacZ] tumors during a 2-dose OVT schedule ($10^6$ PFU) with or without CLL treatment 24 h after the first virus administration. VSV-FLUC was administered as a second dose 48 h after initial treatment with VSV-NR. Results are shown as mean with individual values ($n = 5$); two-way ANOVA. **b** Bioluminescence images for the groups shown in (**a**) 48 h after administration of the second dose of VSV. **c, d** Luminescence intensities for CT26[LacZ] tumors during a 2 dose OVT schedule ($10^6$ PFU VSV-FLUC, 48 h between doses) with or without monocyte depletion by CLL (**c**) or anti-CCR2 antibody (**d**) administered between viral doses. Results are shown as mean with individual values ($n = 5$ (**c**); $n = 7$ (**d**)); two-way ANOVA. **e** Tumor measurements for CT26[LacZ]-bearing mice treated with one or two doses of VSV-FLUC ($10^6$ PFU, 48 h between i.v. injections) ±anti-CCR2. Results are shown as mean ± SEM ($n = 5$); two-way ANOVA followed by Sidak's multiple comparisons test. **f** Kaplan–Meier survival plots for animals treated with one or two doses of VSV-FLUC ± anti-CCR2; log-rank test.

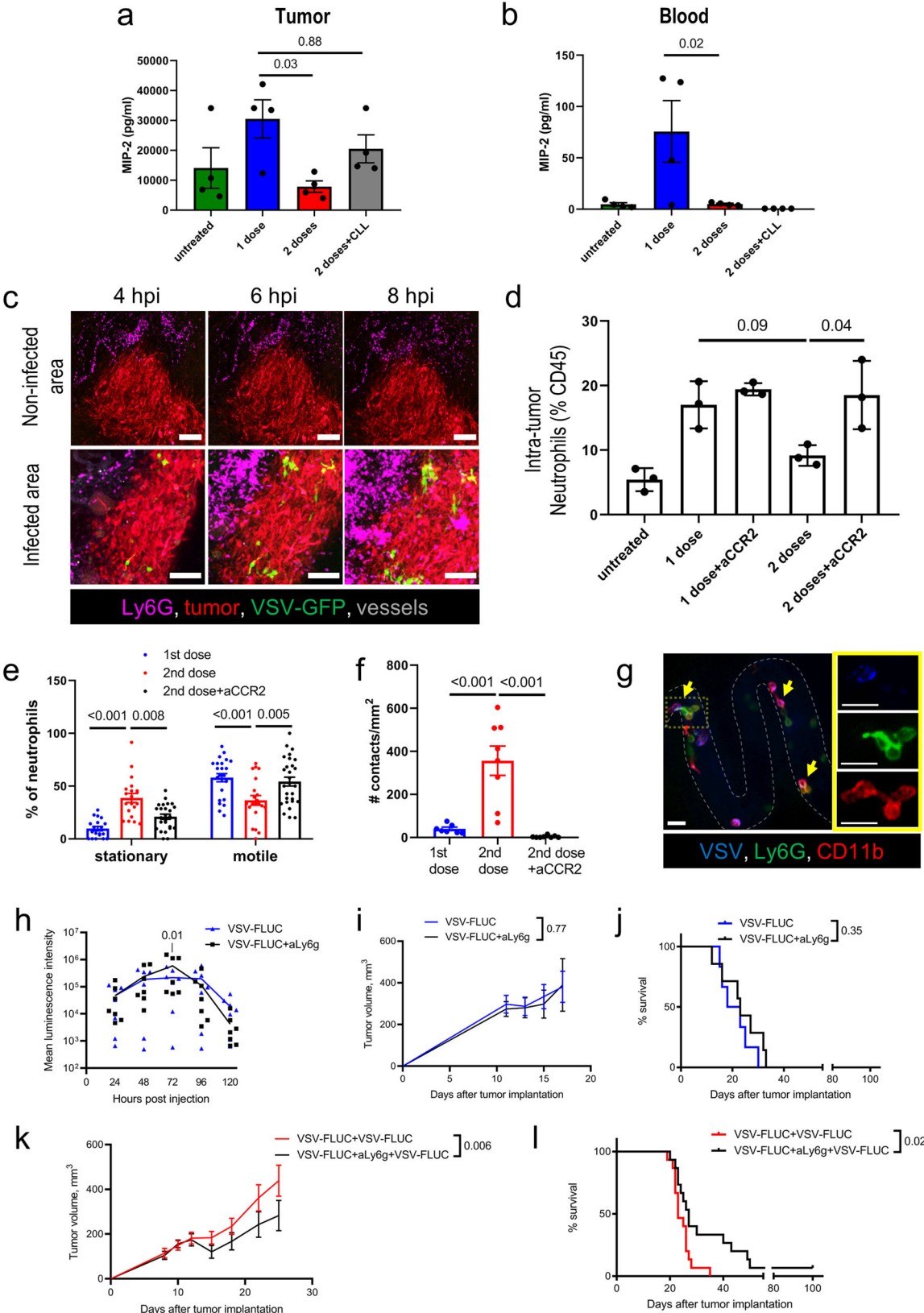

antiviral immunity. This activation of the innate immune response dramatically impacts the infectivity of the second OV dose. Despite reduced infection by the second viral dose, repeated administration of OVT post-conditions the tumor microenvironment for enhanced and sustained infection of the first OV dose, augmenting the anticancer T cell response. Critically, OV-mediated activation of monocytes is central to the antitumor effects of the repeat viral dose. Intravascular monocytes rapidly bind circulating second dose OV and initiate sustained interactions with intravascular neutrophils, resulting in reduced neutrophil infiltration of the tumor. This reduction in intratumoral neutrophils leads to sustained viral infection of the tumor. In

**Fig. 4 OVT limits, in a monocyte-depended fashion, neutrophil antiviral activity within infected tumors.** MIP-2 response in tumor interstitial fluid (**a**) and blood (**b**) to a single or repeated VSV treatment at 56 h following single VSV treatment or 8 h following a second i.v. injection of VSV ($10^6$ PFU) ± CLL 24 h post first virus administration; mean ± SEM; n = 4 (one-way ANOVA followed by Tukey's multiple comparisons test). **c** Time-lapse images of VSV-infected (green) and uninfected regions of a CT26$^{LacZ}$ tumor (red); neutrophils (magenta); vessels (Qtracker 655, gray); scale bar, 100 μm. **d** FC analysis of intratumor neutrophils 72 h following a single VSV dose or 24 h post second VSV dose ($10^6$ PFU) ± 20 μg anti-CCR2 24 h after initial virus administration; mean ± SEM; n = 3 (one-way ANOVA followed by Tukey's multiple comparisons test). **e** IVM analysis of neutrophils behavior 72 h following a single VSV dose or 24 h post second VSV dose ($10^6$ PFU) ± anti-CCR2 24 h after initial virus administration; mean ± SEM (two-way ANOVA followed by Sidak's multiple comparisons test). **f** IVM quantification of neutrophil-monocyte interactions following single and repeat dosing of VSV ± anti-CCR2 treatment; mean ± SEM; n = 8 (one-way ANOVA followed by Tukey's multiple comparisons test). **g** Representative IVM image of neutrophils (green) interacting with virus (blue) bound monocytes (red). The insert shows individual channels for the ROI (dotted yellow box). Scale bar, 25 μm. **h** Luminescence intensity for CT26$^{LacZ}$ tumors following single dose OVT treatment ($10^6$ PFU VSV-FLUC i.v.) ±anti-Ly6g 24 h after virus administration; results are shown as mean with individual values (n = 7); two-way ANOVA followed by Sidak's multiple comparisons test. **i, k** Tumor measurements for CT26$^{LacZ}$-bearing mice treated with one (**i**) or two (**k**) doses of VSV-FLUC ($10^6$ PFU, 48 h between i.v. injections) ± anti-Ly6G 24 h after the first dose of VSV; mean ± SEM (n = 7 for single dose, n = 10 for multiple dose groups); two-way ANOVA. **j, l**. Kaplan-Meier survival plots for mice treated with one (**j**) or two (**l**) doses of VSV-FLUC ± anti-Ly6G (n = 7 for single dose, n = 15 for multiple dose groups); log-rank test.

parallel to this regulation of the inflammatory antiviral response, OV-mediated monocyte activation also enhances CD8 + cell accumulation in the tumor, leading to improved tumor clearance.

Perhaps the most surprising aspect of the mechanisms engaged by repeat OV dosing was the observation that the second viral dose was essentially unable to infect the tumor. This remarkable (>2-log) decrease in tumor infection by the second OV dose is most likely associated with an enhanced systemic antiviral response elicited by the initial VSV$^{ΔM51}$ injection[36,37]. Following first dose administration, systemic shifts in both myeloid and lymphoid cells were observed in the blood and spleen. It has been shown previously that NK cells, NKT cells and granulocytes can infiltrate the infected tumor early, limiting viral spread before generation of adaptive immunity[16–18]. This innate antiviral response involves both the direct killing of infected cells and the production of antiviral cytokines. Importantly, lack of infection by the second dose was not restricted to the tumor, as infection of the spleen or lymph nodes was not seen, a finding that points to the activation of a generalized systemic antiviral response by the first dose of virus and the involvement of host cells beyond the tumor.

Despite lacking infectivity, the second VSV$^{ΔM51}$ dose did enhance and prolong tumor infection resulting from the initial viral dose. These results indicate the repeat OV dose modulates the tumor microenvironment to provide favorable conditions, supporting ongoing viral replication despite the tumor being resistant to new infections. To better understand the mechanisms driving this immune modulation, we tracked the in vivo behavior of VSV$^{ΔM51}$ virions delivered 48 h after the initial dose. Surprisingly, we observed frequent interaction between VSV virions and intravascular CD11b + Ly6c+ monocytes. These cells were positive for CD169 and F4/80, consistent with previous reports on the expression of these cell markers by inflammatory monocytes following viral infection[38–40]. Interestingly, these monocytes were largely absent from the tumor vasculature prior to the administration of the first dose of OV and were only recruited following the initial viral infection of the tumor.

The enhanced ability of monocytes to capture circulating OVs has been reported for other viruses. Similar monocyte response dynamics have been demonstrated after intravenous administration of reovirus[41–43] and monocytes activated by GM-CSF treatment have enhanced OV-capture efficiency due to increased expression of FC-receptors[44,45]. In addition to viral sequestration from the intratumor circulation, it should be noted that viral capture has been reported by CD169 + cells in the spleen and lymph node[22,27,28].

Monocytic cells are known to bind a broad range of viruses, including VSV[34], adenovirus[5,46,47], and oncolytic measles

virus[48,49]. Indeed, in some cases, monocytes are thought to act as vehicles, transferring systemically administered virions to cancer cells[44]. For this reason, ex vivo loading of monocytes, macrophages and MDSCs has been explored to enhance delivery of OVs to the tumor microenvironment; though our data suggests the potential advantage of these virus-monocyte interaction may be less about viral delivery and more about immune modulation. Though viral binding in lymphoid tissues (spleen, lymph nodes) or ex vivo under cell culture conditions is possible, our work reveals that this process can also occur in the peripheral vasculature of the tumor, demonstrating that monocytes are able to capture circulating pathogens under the sheer conditions associated with blood flow. This observation places these intravascular cells in a key position to regulate the local tumor microenvironment in response to a second dose of i.v. administered OV.

A second dose of OV administered 48 h after the initial infection triggers an array of alterations to the systemic and tumor immune landscapes. Following administration of the second viral dose, intravascular monocytes interact with, and bind intravascular neutrophils, forming stable aggregates. The second viral dose also results in reduced production of key inflammatory chemokines such as MIP-2, a potent neutrophil chemoattractant[31] initially induced by the first dose of OV. In agreement with the earlier report[6], our results demonstrate the neutrophils limit virus replication in the tumor, however, we did not observe vascular shutdown after VSV injection described by Breitbach et al. in the same tumor model. The discrepancy could be due to the fact that we used a 500-fold lower VSV dose. Neutrophils are well-equipped to fight invading pathogens, including viruses[50,51]. Direct antiviral effects of neutrophils are attributable to phagocytosis, neutrophil extracellular trap formation, reactive oxygen species production, and proteolytic enzymes release. In particular, human neutrophil peptide 1 inactivates VSV and other viruses in vitro[52]. While the first OV dose resulted in neutrophil infiltration of infected tumor regions, the second OV dose altered the tumor microenvironment, leading to reduced neutrophil infiltration, establishing conditions necessary to enhance and prolong viral replication. These immune alterations also impact adaptive immunity, triggering a monocyte-dependent enhancement of the CD8 + T cell response and tumor clearance.

The multiple mechanisms of immune modulation facilitated by monocytes following the second dose of OV create an interesting paradox; whereas neutrophil infiltration limits viral replication and overall therapeutic efficacy, enhanced CD8 + T cell responses, which are also antiviral, are beneficial. This apparent discrepancy is likely explained by the timing of the responses – early inhibition of the viral infection is detrimental, attenuating the

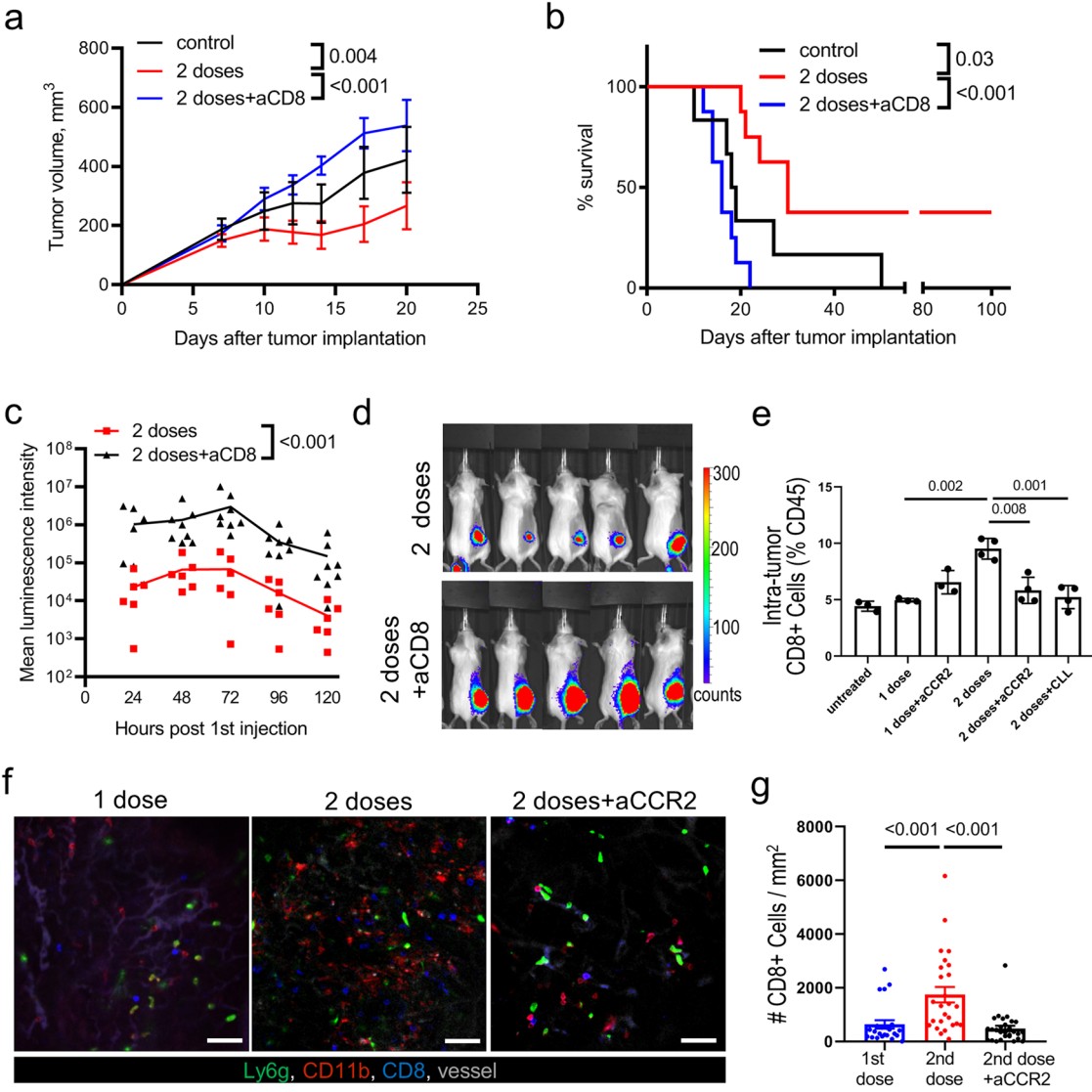

**Fig. 5 OV-monocyte interactions promote CD8 + T cell recruitment. a** Tumor measurements for untreated CT26^LacZ-bearing animals and mice treated with two doses of VSV-FLUC (10^6 PFU, 48 h between i.v. injections) ±anti-CD8 treatment. Results are shown as mean ± SEM ($n = 6$ for untreated group; $n = 8$ for VSV-treated groups); two-way ANOVA followed by Sidak's multiple comparisons test. **b** Kaplan-Meier survival plots for the groups shown in **a**; log-rank test. **c** Luminescence intensity for CT26^LacZ tumors following a 2-dose OVT treatment schedule (10^6 PFU VSV-FLUC, 48 h between doses) ± CD8 depletion. Results are shown as mean with individual values ($n = 6$); two-way ANOVA. **d** Representative bioluminescent images for groups shown in **c** 24 h post second virus injection. **e** FC analysis of tumor CD8 + cells 72 h following a single VSV dose or 24 h post second VSV dose (10^6 PFU) ± anti-CCR2 or CLL treatment 24 h after initial virus administration. Results are shown as percentage of CD45 + cells and plotted as mean ± SEM (one-way ANOVA followed by Tukey's multiple comparisons test). **f** Representative IVM images of tumor vessels and leukocytes at 72 h following a single VSV dose or 24 h post second VSV dose ± anti-CCR2 treatment 24 h after initial virus administration. Green, neutrophils; red, monocytes; blue, CD8 + cells. Vessels counterstained by Qtracker 655 (gray); scale bar, 50 μm. **g** IVM analysis of CD8 + cells in CT26^LacZ tumors 72 h following a single VSV dose or 24 h post second VSV dose (10^6 PFU) ± anti-CCR2 treatment 24 h after initial virus administration. Results are shown as CD8 + cell counts per mm^2 and plotted as mean ± SEM (one-way ANOVA followed by Tukey's multiple comparisons test).

ensuing antitumor response, whereas later clearance of infection by recruited CD8 + T cells does not negatively impact tumor clearance as by this time the virus has done its job – the antitumor response has been initiated, lymphocytes have been activated, and cellular recruitment is underway. At this later time point, viral infection is no longer critical for tumor clearance. Moreover, these multiple mechanisms identify two distinct, nonoverlapping roles for OV; (1) direct infection of cells in the tumor triggering localized inflammation, focusing the host response, and (2) reprogramming of recruited leukocytes leading to sustained viral infection and enhanced tumor clearance. It remains to be determined how long spacing between doses can be

extended while continuing to yield functional retrograde reprogramming of the tumor microenvironment. This temporal gap is likely to be related to the specific viral strain, route of administration, and therapeutic dose and will require individual optimization.

One limitation of the current work is using immunologically hot tumor models (CT-26^LacZ and M3-9-M). Further studies are needed to reveal if the described interplay between the consequent doses is relevant to highly immunosuppressed tumors; however, our findings also have implications beyond the development of optimized OVT regimens as viral vectors have received much attention as potential vaccine platforms. The development

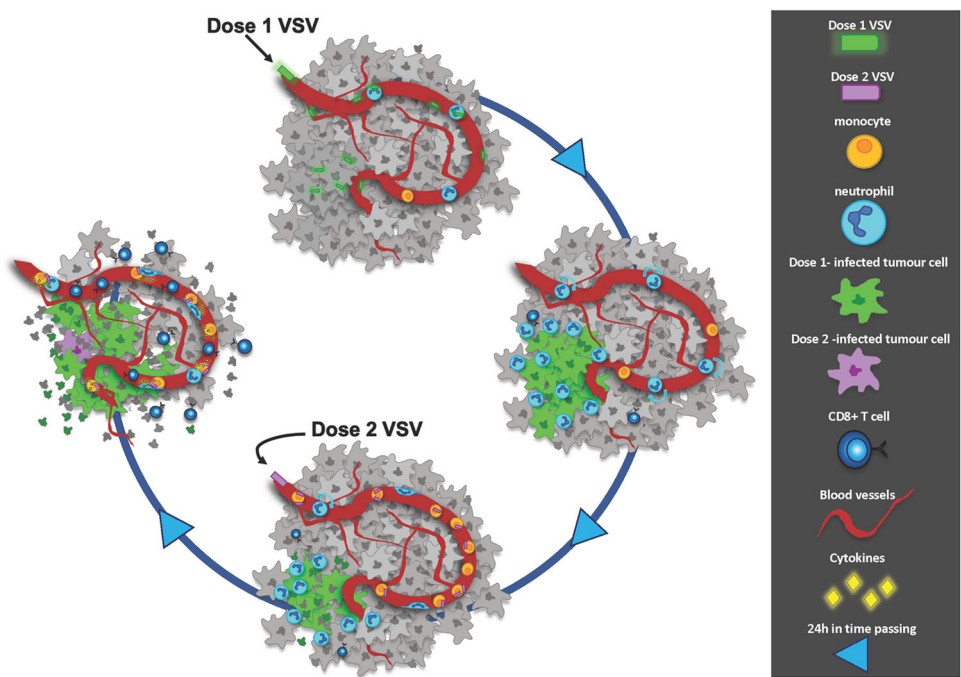

**Fig. 6 Proposed mechanisms of tumor clearance in facilitated by a multidose OVT protocol.** Following the first dose of OV, virus is observed binding to the endothelium and neutrophils intravascularly, and tumor cells within the tissue initiating tumor cell infection. Once foci of infection have been generated recruitment of neutrophils to the foci is observed. Over the course of the next 24 h the neutrophils limit the tumor cell infection. Monocytes become abundant in the tumor vasculature and following a second dose of VSV are frequently observed capturing virus particles. This second dose of virus leads to enhanced infection from the first dose of virus by modulating the tumor environment in a monocyte-dependent fashion. Virus bound monocytes interact with neutrophils within the vessels, the cytokine profile within the tumor is altered, neutrophil recruitment is reduced, and CD8 + T cell numbers increased. Oncolysis of the tumor cells from the enhanced first dose, in addition to increased numbers of CD8 + T cells 24 h following the second dose synergize to result in more effective tumor clearance compared to a single dose of OV.

of multiple adenovirus-based SARS-CoV-2 vaccines[53,54] and the inclusion of VSV-based platforms in Phase III clinical trials[55–57] highlight the need to understand the multiple levels of interaction between the vector and the immune system beyond antigen expression and development of a cognate immune response. Though some platforms utilize a classic prime-boost strategy, the identification of distinct roles for first and second doses administered days apart may help further enhance host immunity. Delivery of a target antigen by an initial dose could be enhanced by rapid (48 h) administration of a second viral dose, leading to sustained antigen expression and better priming of the host T cell response. This approach may have broad application as vaccine strategies shift from humoral-immunity based approaches to newer vaccines that strive to engage better cellular immunity. It remains to be determined if enhancement of the initial viral dose can better support adaptive immune development and long-lasting immune memory, though the potential is exciting.

To summarize, primary and secondary doses of OV are not simply additive but rather engage two distinct host responses that, working together, yield improved therapeutic efficacy, tumor clearance and animal survival. Whereas the first viral dose mediates its effects through infection, the second mediates its effects through modulation of the tumor microenvironment, supporting and extending viral replication originating from the initial dose of OV, and enhancing CD8 + T cell immunity. The first OV dose infects cells in the tumor, driving inflammation and facilitating cell recruitment. Central to this response is the accumulation of intravascular monocytes. Following this initial dose, monocytes adhere within the tumor vasculature, ideally positioned to bind subsequent OV doses, acting as central players in modulating the host immune response. Depletion of these monocytes attenuates the effect of the second OV dose, limiting

viral replication in the tumor and resulting in a suboptimal CD8 + T cell response. Understanding these differential roles of the first and second viral dose will allow us to better optimize a multidose OVT treatment regimen. Overall, although subsequent doses of OV appear to fail to infect cancer cells, through activation of monocytes, a second dose of OV can act as a key piece of OVT treatment regimens. A further understanding of these distinct first and second dose immune responses and the cross-talk between virus, innate, and adaptive immunity will help improve the existing protocols of OVT and its combination with other cancer treatments.

## Methods

**Animals.** Eight-week-old female BALB/c and C57bl/6 mice were purchased from Charles River Laboratories (Wilmington, MA) and maintained in specific-pathogen free facilities at the University of Calgary. At the time of use, animals were between 8 and 10 weeks of age and weighed 20–25 g. All experiments involving animals were approved by the University of Calgary Animal Care Committee (Protocol #AC15-0081) and conform to the guidelines established by the Canadian Council for Animal Care.

**Viruses and cells.** Vesicular stomatitis virus (VSV$^{\Delta M51}$), VSV expressing firefly luciferase (VSV$^{\Delta M51}$-FLUC) and VSV expressing green fluorescent protein (VSV$^{\Delta M51}$-GFP) were originally generated by Dr. J. Bell (Children's Hospital of Eastern Ontario) and kindly provided by Dr. X. Lun (University of Calgary). Maraba virus (MG1) was kindly provided by Dr. D.Stojdl (CHEO). These strains were propagated in monolayer cultures of Vero cells, the supernatant harvested at 50% cytopathic effect. After removing cellular debris by centrifugation (300 × g for 5 min at 4 °C) and passing the virus-containing supernatant through a 0.2 μm filter, the clarified supernatant was centrifuged at 28,000 × g for 1.5 h at 4 °C. The virus-containing pellet was resuspended (1 mM EDTA, 1 mM NaCl, 1 mM Tris, pH 7.4) and in turn centrifuged through an Optiprep gradient at 160,000 × g for 1.5 h at 4 °C. A single band of concentrated virus particles was collected, aliquoted, titered by plaque assay on Vero cells, and stored at −80 °C[26,58].

CT-26 LacZ cells, a bioevolved version of the ATCC CT-26.CL25 cell line that has acquired deficiencies in type I interferon responsiveness and is hypersensitive

to VSV$^{\Delta M51}$ (CT-26$^{LacZ}$; murine colon adenocarcinoma[59]) were obtained from Dr. J. Bell. M3-9-M cells were obtained from Dr. C. MacKall (Stanford). Both cell lines were cultured in RPMI supplemented with 10% fetal calf serum. CT-26 $^{LacZ}$-RFP for IVM was generated by lentivirus transduction.

**Labeling virus**. Alexa Fluor 647 succinimidyl esters (AF647SE, Molecular Probes, Invitrogen) were reconstituted in DMSO and 5 μl of Alexa dye (100 μg/ml prepared in phosphate buffered saline [PBS]) was added to 45 μL of VSV$^{\Delta M51}$ (5 × 10$^{10}$ PFU/ml) to get a final concentration of 10 μg/ml of dye, while stirring gently. Virus was incubated with the dye for 20 min at room temperature with gentle inversions every 5–10 min. To remove excess unbound dye, the labeling mixture was transferred to Amicon Ultra-4 Centrifugal Filter Units (100 kDa membrane; EMD Millipore) and washed twice in 1 ml of PBS by centrifuging (4000 × $g$) for 10 min at 4 °C. After incubation, labeled samples of virus were titered using TCID$_{50}$ assay.

**TCID$_{50}$ assays**. To determine viral titers, 10$^4$ Vero cells/well were seeded in 96-well plates and cultured in 10% fetal bovine serum (FBS) DMEM until 80–90 % confluence. CT-26$^{LacZ}$ tumors were extracted 72/24 h post first/second VSV injections and immediately frozen in liquid nitrogen. The next day the thawed tumors were weighted, 1 mL of serum free media was added and samples were homogenized in sterile conditions, centrifuged for 5 minutes at 3000 × $g$ to separate tumor infiltrated fluid (TIF) from debris. TIF samples were assessed as 10-fold serial dilutions ranging from 10$^{-1}$ to 10$^{-6}$ in serum free DMEM. Dilutions of TIF were added to the Vero cells (4 wells for each dilution) and incubated for 1 h. Cells were washed and fed with growth media containing antibiotics and cultured at 37 °C with 5% CO2. After 72 h the TCID$_{50}$ was calculated by counting the number of wells with GFP expression based on the Reed and Muench method.

**Antibodies, stains, and treatments**. BV-421-conjugated rat anti-mouse Ly6g (clone 1A8, 0.2 mg/ml), FITC-conjugated rat anti-mouse CD11b (M1/70, 0.5 mg/ml), rat anti-mouse CD16/CD32 (Fc block, clone 2.4G2, 0.5 mg/ml) were purchased from BD Biosciences Pharmingen (San Diego, CA). FITC-conjugated rat anti-mouse Ly6g (1A8, 0.5 mg/ml), AF488-conjugated rat anti-mouse Ly6c (HK 1.4, 0.5 mg/ml), PE-conjugated rat anti-mouse CD169 (3D6.112, 0.2 mg/ml), PE-conjugated rat anti-mouse F4/80 (BM8, 0.2 mg/ml), PE-conjugated rat anti-mouse CD11b (M1/70, 0.2 mg/ml), PE/Cy7-conjugated hamster anti-mouse CD11c (N418, 0.2 mg/ml), PE-conjugated rat anti-mouse CD8b (YTS156.7.7, 0.2 mg/ml), PerCP-conjugated rat anti-mouse Ly6c (HK 1.4, 0.2 mg/ml), PerCP-conjugated rat anti-mouse CD4 (GK1.5, 0.2 mg/ml), PE-conjugated rat anti-mouse CD19 (1D3, 0.2 mg/ml), FITC-conjugated rat anti-mouse CD45R/B220 (RA3-6B2, 0.5 mg/ml), APC-conjugated rat anti-mouse Ly6g (1A8, 0.2 mg/ml), FITC-conjugated rat anti-mouse F4/80 (BM8, 0.5 mg/ml), APC/Cy7-conjugated rat anti-mouse CD45 (30-F11, 0.2 mg/ml), PE-conjugated rat anti-mouse Ly6g and Ly6c (R6B-8C5, 0.2 mg/ml), PerCP-conjugated rat anti-mouse Ly6g and Ly6c (R6B-8C5, 0.2 mg/ml) were purchased from Biolegend (San Diego, CA). Rat anti-mouse CD8a eFluor® 660 (53-6.7, 0.2 mg/ml), PE/Cy7-conjugated rat anti-mouse CD11b (M1/70, 0.2 mg/ml) were purchased from eBioscience (San Diego, CA). For cellular depletion, rat anti-mouse Ly6g (1A8, 8.17 mg/ml), rat anti-mouse CD8 (YTS 169.4, 8.05 mg/ml) were purchased from BioXcell (West Lebanon, NH). Rat anti-mouse CCR2 (MC21, 1 mg/ml) antibody was kindly provided by Dr. M. Mack (University of Regensburg). Clodronate containing and empty liposomes were a kind gift of Dr. F. Jirik (University of Calgary). To ensure specificity of antibodies in vivo, isotype controls were also purchased and used to ensure there was no Fc-mediated labeling of cellular targets. Qtracker® 655 Vascular Labels (Invitrogen, 2 μM) and FITC-conjugated bovine serum albumin (BSA; Sigma Aldrich, 5 mg/ml) were used in IVM experiments for counterstaining vessels.

**Tumor model and animal treatment**. Syngeneic tumors were established by s.c. injection of 10$^6$ cells in 50-μl into the right hind flank. Colon cancer cells (CT-26 $^{LacZ}$ or CT-26 $^{LacZ}$-RFP) were implanted into BALB/c mice, murine rhabdomyo-sarcoma (M3-9-M) were implanted into C57bl/6 mice. On day 10-11 post-implantation, mice were treated by intravenous injection of VSV$^{\Delta M51}$ (1 × 10$^6$ PFU or 5×10$^8$ PFU) with or without a second dose 48 h after the primary treatment. In some experiments, animals additionally received a third dose 48 h after a second one. To determine the specific contribution of the initial VSV dose, animals were administered VSV$^{\Delta M51}$-FLUC/GFP (10$^6$ PFU OV for animals bearing CT-26$^{LacZ}$ tumors; 5 × 10$^8$ PFU for animals bearing M3-9-M tumors) followed by second injection of VSV$^{\Delta M51}$ without reporter transgenes (VSV$^{\Delta M51}$-NR; Supplementary Fig. 6a). To assess the contribution of the second dose the reverse order of viruses was applied - VSV$^{\Delta M51}$-NR as an initial treatment and VSV$^{\Delta M51}$-FLUC/GFP as the second dose (Supplementary Fig. 6b). Control animals were treated with PBS by i.v. injection. Flank tumor diameters were measured every other day and tumor volume was calculated using the equation: V = ½ a$^2$ × b, where a – is the smaller of two orthogonal measurements. The mice were euthanized when tumors reached 500 mm$^3$ in size. Monocyte depletion was achieved by intraperitoneal injection (i.p.) of 1 mg/kg of CCR2-specific antibodies (MC-21) or with 50 mg/kg clodronate liposomes i.v (Supplementary Fig. 6c). Other cell populations were depleted by i.p. administration of antibodies (neutrophils, single administration of 10 mg/kg anti-Ly6g (1A8) (Supplementary Fig. 6d); CD8, 12.5 mg/kg anti-CD8 (YTS 169.4) administered on day 8 post tumor implantation followed by injections of 5 mg/kg on days 11, 15, 22 post-infection (pi) (Supplementary Fig. 6e).

**Intravital microscopy**. For all experiments, mice were anesthetized by i.p. injection of 200 mg/kg ketamine (Bayer Inc Animal Health, Toronto, Ontario, Canada) and 10 mg/kg xylazine (Bimeda-MTC, Cambridge, Ontario, Canada). The tail vein was cannulated to permit the delivery of fluorescently labeled antibodies (0.03 mg/kg CD11b-FITC; 0.03 mg/kg Ly6G-BV421; 0.125 mg/kg Ly6c-AF488; 0.07 mg/kg CD169-PE; 0.125 mg/kg F4/80-FITC; 0.125 mg/kg CD8b-PE) and for maintenance of anesthetic. Mouse body temperature was maintained using a heated stage. For s.c. tumor preparation, midline incision along the spine was made and skin reflected. The thin connective tissue membrane overlaying the inside surface of the skin was removed and edges of the skin flap were secured by sutures to expose and stabilize the tumor for imaging[60].

Intravital resonant-scanning confocal and multiphoton microscopy was performed using a Leica SP8 inverted microscope (Leica Microsystems, Concord, Ontario, Canada), equipped with 405-, 488-, 552-, and 638-nm excitation lasers, 8 kHz tandem scan head and spectral detectors (conventional PMT and hybrid HyD detectors) for superficial imaging (up to 100 μm). Additionally, a tunable multiphoton excitation laser (700-1040 nm - Newport Corporation, Irvine, CA) and external PMT detectors (Leica) was used for deeper imaging of tumors (up to 300 μm).

Cell behavior was classified as previously described[61]. A cell was considered either: (a) crawling, if it maintained continual interaction with the vessels and traveled a distance greater than its own diameter within 5 min, or; (b) stationary, if the cell did not travel more than one cell diameter. Interaction between cells was noted if two cells were in direct contact for more than 5 min. Quantification of cell numbers and interactions was performed manually in 5–10 FOV for each animal and normalized to the area (mm$^2$) of visible tumor tissue.

**Flow cytometry**. Blood was obtained through cardiac puncture and collected in syringes containing 100 U heparin; red blood cells were lysed using Ammonium-Chloride-Potassium (ACK; Gibco). Tumors, spleen, and lymph nodes were harvested from euthanized animals, placed in ice-cold PBS, and homogenized by mechanical disruption. Tumor samples were additionally treated with 1 mg/ml collagenase I and 0.1 mg/ml DNase I for 30 min at 37 °C and single-cell suspensions were generated using a GentleMACS (Miltenyi Biotec) followed by passage through a 70-μm nylon mesh. Following 3 washes in cold PBS, cells were blocked with anti-CD16/CD32 mAbs (1/100) in FACS wash buffer (FWB; PBS, 2 % FBS, 5 mM EDTA) for 30 min at 4 °C, followed by staining with fluorophore-conjugated antibodies (1/100) in FWB for 30 min at 4 °C. Cells were washed 3 times in FWB and analyzed on an Attune Acoustic Focusing Cytometer (Life Technologies). Figures were generated using either FlowJo (Tree Star) or Attune Cytometer software. The distinct cell populations were gated as shown in Supplementary Fig. 7. In some experiments 10 μg of APC/Cy7-conjugated anti-CD45 antibodies were i.v. injected 10 min prior to harvesting tumors for labeling intravascular leukocytes.

**Bioluminescence imaging**. At the indicated time points, mice infected with VSV$^{\Delta M51}$-FLUC were injected i.p with 150 mg/kg of firefly D-luciferin (Gold Biotechnology) in PBS and allowed to rest for 10 min. Imaging was conducted using IVIS imaging system series 100 (Xenogen, Alameda, CA, USA) and photon emission values were calculated with Living Image v2.5 software (Xenogen).

**Cytokine analysis**. To obtain TIF, 0.1–0.3 g of fresh tumor tissues were cut into small pieces (1–3 mm$^3$), processed in homogenizer for 30–60 s and then placed in a 15-mL conical plastic tube containing cold PBS (1 mL PBS per 0.25 g tumor tissue). Samples were centrifuged at 100 x $g$ for 3 min and the supernatants were transferred to new microtubes. Samples were further centrifuged at 2500 × $g$ for 20 min at 4 °C. To obtain serum, mice were anesthetized with a ketamine and xylazine cocktail (100 mg/kg and 10 mg/kg, respectively) and blood was drawn by cardiac puncture. Whole blood was clotted for 30–60 minutes then centrifuged at 2000 × $g$ for 20 min. TIF and sera supernatants were analyzed using a MILLIPLEX MAP Mouse Cytokine/Chemokine Panel as per manufacture's instructions (EMD Millipore).

**Statistics and reproducibility**. Statistical analyses were performed using unpaired Student's $t$ test, Mann–Whitney test, 1-way or 2-way ANOVA with Multiple Comparisons test, and Kaplan–Meier assessment (GraphPad Prism 8.0). Most experiments were replicated one–two times. For animal experiments, the sample size was generally set to $n$ = 4–7 per group. The $n$ value is defined within each figure and/or legend. All values and variances were generated from biological replicates.

**Reporting summary**. Further information on research design is available in the Nature Portfolio Reporting Summary linked to this article.

## Data availability
The data that support the findings of this study are available within the article and its Supplementary Information and Supplementary Data 1 or from the corresponding authors on reasonable request.

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

## Acknowledgements

The authors thank Dr. D.Stojdl (CHEO), Dr. J. Bell (OHRI), and Dr. C. MacKall (Stanford) for providing OV strains and cell lines. This work was supported by Canadian Foundation for Innovation, Canadian Cancer Society (Mahoney and Jenne, Grant#DF-18-5), and CIHR (Mahoney and Jenne, Grant#420828). CNJ is supported by the Canada Research Chairs Program; MT is supported by Natural Sciences and Engineering Council of Canada, JR is supported by Vanier Canada Graduate Scholarship. The authors also thank the Alberta Children's Hospital Foundation and Women in Insurance Cancer Crusade for generous support.

## Author contributions

V.N. - methodology, investigation, validation, and writing – original draft preparation; J.R. - validation and writing – review & editing preparation; M.T. - investigation, validation, and writing – review & editing preparation; C.Z. – investigation; M.Tse – investigation; R.P.D. – investigation; D.K. – investigation; A.R. – investigation; H.D. - investigation; S.V. – investigation; L.K.M. – investigation; E.K.K. – investigation; V.P.Ch - writing – review and editing preparation; D.J.M. - conceptualization, supervision, and writing – review & editing preparation; C.N.J. - conceptualization, supervision, writing – review & editing preparation, project administration.

## Competing interests

The authors declare no competing interests.
