## [Peer Review File · Communications Biology]

Reviewers' comments:

Reviewer #1 (Remarks to the Author):

The manuscript by Naumenko et al., titled "Repeat dosing improves oncolytic virus therapy via interactions with intravascular monocytes", tackles the question of whether repeat dosing is beneficial during oncolytic virus (OV) therapy. This is a very important question in the field that has not been thoroughly addressed and the findings could help guide the design of dosing regimens during OV therapy in patients.

The authors found that 2 doses of the OV VSVd51 administered 48h apart resulted in more virus replication within the tumour compared to a single dose, leading to better efficacy. Surprisingly, they show that the second dose of virus does not directly infect the tumour cells. Rather, the second dose is sequestered by monocytes/macrophages that are recruited to the tumour after the first dose, and this results in changes in the tumour microenvironment that ultimately lead to fewer neutrophils, which normally limit virus replication in the tumour, being recruited to the tumour in a monocyte dependent fashion.

Overall, the manuscript is well-written with the text being clear and concise which makes it easy to read. The figures are also well-presented for the most part, although I have added a few suggestions for potential improvements in my comments. The experiments performed are well designed and the techniques used are appropriate to answer the questions at hand. Notably, the authors are experts in intravital microscopy and have previously developed tools to study the interaction between virus particles and cells of the tumour microenvironment, which brings a unique analytical tool to study OVs. The major findings were confirmed in 2 separate tumour models (CT26 LacZ and M3-9-M) using VSVd51 for the majority of experiments, but also confirming key findings with the clinical candidate OV MG1. Overall, the conclusions are largely supported by the data presented with no major experimental flaws.

General comments:

1) The findings of this study may be specific to VSV therapy. Different viruses may interact with different cell types, have different replication kinetics, may express immunomodulatory proteins etc. Therefore, the findings cannot be generalized to OVs in general. I am not suggesting repeating key experiments with other OVs (although that would be great), but the title should be changed to specify VSV.

2) I think it is very informative to show bar graphs with individual data points overlaid.

3) Some of the bar graphs of the flow cytometry data are confusing. The Y axis should be better labeled in a number of figures. For example, in Fig S5E, I think the bar graph shows % CD8 T cells gated on CD45+ cells based on the text. This is not clear on the graph or in the figure legends. It actually looks like the data is showing the % CD45+ cells in each condition. Consider changing the Y axis to % CD8+ (of CD45+ cells) (as is done in Fig. 5E). This applies to a few of the flow figures in the paper, including S4A.

4) It would be nice to include some representative flow plots in the supplemental data to show gating strategies etc.

Specific comments:

5) In the introduction, the authors describe commonly used dosing schedules for a number of OVs, including Maraba virus, a rhabdovirus closely related to VSV. It is noted that Maraba is given as 2-4 doses, with each dose 3-4 days apart. Could the authors recapitulate this dosing schedule in their

experimental model. Specifically, what happens if you increase the time between the first and second dose to 3 days or 5 days, do you still see enhancement by the second dose? Similarly, what happens if you give a 3rd dose 48h after the 2nd dose, do you still see enhancement?

6) I appreciate why the authors used suboptimal virus doses (1e6 PFU) in the CT26-LacZ model given that it is relatively easy to achieve cures in this tumour model with the standard dose (5e8 PFU). However, it has been shown that there is a threshold dose of virus needed to overcome the barriers to systemic delivery of OVs. If the second dose is increased to 5e8 PFU, would this overcome the antiviral activity of the neutrophils present in the tumour microenvironment and lead to tumour infection with the second dose, and still be enhanced by monocytes?

7) How does the 2nd dose of VSV lower MIP-2 levels? What cells are making MIP-2? I would have thought macrophages/monocytes, but depletion of these cells with CLL increases MIP-2 levels, suggesting it is not the case. Also, can you block MIP-2 with neutralizing antibodies to show that the chemokine is critical for the enhancement effect of the 2nd dose?

8) The authors demonstrate that neutrophils limit virus replication in both the 1st and 2nd dose, suggesting that these cells have antiviral activity. How do neutrophils limit VSV replication in this model, especially given that CT26-LacZ cells are interferon non-responsive? It does not seem to be via phagocytosis of virus particles as less virus associates with neutrophils in the 2nd dose compared to the 1st dose. Do they induce an IFN-independent antiviral state in tumour cells? Are neutrophil extracellular traps sequestering virus? Does the interaction between monocytes and neutrophils change gene expression of the neutrophils? Additional experiments aimed at answering this question would provide mechanistic insights on the role of neutrophils during OV therapy.

9) It has been shown that neutrophils mediate vascular shutdown following VSV infection in the CT26-LacZ model (PMID: 17579581). Can you rule out that this is not the reason there are fewer neutrophils in the tumour bed? If not, this should be elaborated in the discussion.

10) The finding that CD8 depletion improves VSV replication is quite interesting. Can you assess VSV binding to CD8 T cells to determine if CD8 T cells do indeed exert antiviral activity against VSV as opposed to simply binding and sequestering virus particles away from the tumour. The difference is there even 24h after the first dose, and at this point the T cells are just becoming activated and finished blasting, so I am not too sure how much effector functions they have this early on after infection.

Minor points:

11) Please verify the p-value in Figure S2B? The CD11b+ Ly6G hi group seems like it should be significant based on the 2 previous groups.

12) I don't see the value in Figure 1D as it is, it simply shows that VSV replicated in the tumour which you have already established. A more appropriate figure would be to compare 1 dose of VSV-GFP, to 2 doses with 1st dose being VSV-NR and 2nd dose VSV-GFP.

13) The y-axis in Fig 4D seems cut-off by the y-axis title.

14) Normally in an uninfected mouse there are more CD4 T cells than CD8 T cells (the ratio is somewhere around 3:2 CD4:CD8), but in Figure S2C and S2E the percentage of CD4 T cells is extremely low (less than 1% CD4 T cells, but 10-15% CD8 T cells). This experiment would have to be repeated as I doubt this is truly the case.

Reviewer #2 (Remarks to the Author):

In this study the authors investigate the fate of multiple doses of oncolytic VSV in the susceptible tumour model, CT-26LacZ. In this study the authors lower the therapeutic dose of VSV in order to investigate the effect of VSV in a multiple treatment regimen. This study provides a novel insight in to the fate of both the first and second dose of VSV by administering VSV-FLUC either as the first or second dose. Moreover, the authors use fluorescently tagged VSV in order to monitor the viruses cellular interactions using IVM. This study shows that the second dose of VSV instead of infecting tumour cells is captured by Ly6C+ cells and that this cell population is critical for therapy in this regimen. The manuscript would of greatly benefited from further investigation of these cells, however i understand that these studies may be beyond the remit of this paper.

The figures in this manuscript are presented clearly and are easy to follow, but it would be beneficial to the reader if the authors could include the gating strategy for the FACS data, and clarify what they have termed neutrophils in figure 4.

Furthermore in Figure 4 A MIP-2 should be presented as a concentration not fold increase.

Could the authors also explain the discrepancy in the survival experiment in 4L as the two doses do not appear to match previously presented survival curves in Figure 1B, 3F and 5B (it would be also helpful if axis were kept consistent throughout the manuscript).

The authors generally describe their findings in a concise and accurate manner however there is one incidence where the sentence is a bit misleading. Line 202 'neutrophils are absent from the tumour 24hrs following the second dose' they are not absent but just return to untreated controls (also the axis related to this figure has been cut off).

In the discussion the authors fail to discuss that the model that they have used is immunologically 'hot' and therefore priming T cells response is possible following OVT, they focus instead more on the onolytic nature despite their results showing that even if you do enhance replication that does not lead to better outcomes (figure 3F vs Figure 4J). Do the authors have anymore information on how this multiple dose regimen performs in resistant or 'cold' tumours?

Reviewer #3 (Remarks to the Author):

The authors have investigated whether repeat intravenous dosing with an oncolytic VSV can enhance efficacy and by what mechanism. They use intravital microscopy to show that the first VSV dose infects neoplastic cells in the tumor whereas the second dose, given 48 hours after the first dose, does not. However, the second dose does lead to enhanced intratumoral spread of the first dose and improved efficacy outcomes in VSV-sensitive tumor models. They further investigate the mechanism whereby the second dose of VSV is able to enhance the intratumoral spread of the first dose and their data suggests it is a consequence of the virus infecting monocytes that have by that time accumulated in tumor neovessels. This somehow modulates the tumor microenvironment leading to enhanced T cell function which ultimately impacts the depth of tumor response.

The central finding of the paper is novel and is of importance for the field of oncolytic virotherapy. Despite the interesting correlateive analyses reported in the paper, the mechanism by which a second dose of virus can enhance the intratumoral spread of a first dose remains something of a mystery,

although it does seem to require that the virus interacts with monocytes that have accumulated on tumor neovessels as a consequence of the infection of tumor cells mediated by the first virus dose.

The studies are detailed and are convincing and the conclusions are likely to have a significant impact for the field. Additional mechanistic studies will be required to further investigate and elucidate the mechanism of action that has been uncovered here, but such studies are probably beyond the scope of the paper.

One suggestion for improvement is that the accompanying videos are difficult to follow and the authors should make an attempt to better annotate them. It is also suggested that they should slow the speed of video replay to allow better tracking of individual cells and viruses. That said, the intravital microscopy videos are an appealing aspect of the paper.

Reviewer #4 (Remarks to the Author):

The manuscript entitled: Repeat dosing improves oncolytic virus therapy via interactions with intravascular monocytes by Naumenko et al. depict the infection dynamics of the oncolytic virus VSV and how a second dose drives an enhanced virus infection and response through the accumulation of monocytes and ultimately CD8+ T cells. With the use of fluorescently labeled viruses, the authors illustrated that it was not the replication of the second dose that mitigated the response, rather it was the influx of monocytes with a decrease neutrophil response that propagated the first dose of OV infection.

Overall, the manuscript provided insight into the benefits of sequential dosing of OVs and that the immune response immediately following OVs is a key for a beneficial therapeutic response.

Below are my specific comments:

Concerns/questions:

- 1) The authors conducted a TCID50 in figure 1 but did not follow up on the virus fitness through the various experiments that manipulated the immune component (i.e., anti-CCR2, or anti-Ly6G). The addition of this would significantly strengthen the author's claims.
- 2) Does a 2nd dose increase monocyte function or accumulation in comparison to a single dosage? It would be important to see if monocyte accumulation is greater or cytokine production is higher in 2nd dose monocytes.
- 3) For flow cytometry analysis a representative gating strategy is essential for determining whether the immune cells are properly identified. This is particularly important for markers like CD169+ Marginal metallophilic macrophages in the spleen which are notorious for not being captured by flow cytometry.
- 4) Do the authors observe a difference in Red pulp macrophages and marginal metallophilic macrophages?
- 5) Please give a rationale for the depletion of CD8+ T cells prior to OV treatment. Also, what is the frequency of CD8+ T cells in nontreated tumors? Does a single dose also have a benefit?
- 6) Do the CD8+ T cells have different functional aspects following a single or 2 dose regimen of OV?

Minor conflict:

1)Would recommend changing the title to Repeated dosing improves...

Referee #1

1. The findings of this study may be specific to VSV therapy. Different viruses may interact with different cell types, have different replication kinetics, may express immunomodulatory proteins etc. Therefore, the findings cannot be generalized to OV_s in general. I am not suggesting repeating key experiments with other OV_s (although that would be great), but the title should be changed to specify VSV –

*We agree with the Reviewer's comment. The title has been changed to "Repeated dosing improves oncolytic **rhabdovirus** therapy via interactions with intravascular monocytes"*

2. I think it is very informative to show bar graphs with individual data points overlaid.

Following the Reviewer's suggestion, we have modified bar graphs to show individual data points for the main figures.

3. Some of the bar graphs of the flow cytometry data are confusing. The Y axis should be better labeled in a number of figures. For example, in Fig S5E, I think the bar graph shows % CD8 T cells gated on CD45+ cells based on the text. This is not clear on the graph or in the figure legends. It actually looks like the data is showing the % CD45+ cells in each condition. Consider changing the Y axis to % CD8+ (of CD45+ cells) (as is done in Fig. 5E). This applies to a few of the flow figures in the paper, including S4A.

The corresponding changes have been made.

4. It would be nice to include some representative flow plots in the supplemental data to show gating strategies etc

Following the Reviewer's suggestion, we have added new Figure S7 to show gating strategy for FC

5. In the introduction, the authors describe commonly used dosing schedules for a number of OV, including Maraba virus, a rhabdovirus closely related to VSV. It is noted that Maraba is given as 2-4 doses, with each dose 3-4 days apart. Could the authors recapitulate this dosing schedule in their experimental model. Specifically, what happens if you increase the time between the first and second

dose to 3 days or 5 days, do you still see enhancement by the second dose? Similarly, what happens if you give a 3rd dose 48h after the 2nd dose, do you still see enhancement?

To address this important point, we tested if the third dose could further enhance the infection. However, we did not find any difference in tumor bioluminescence between mice treated with two or three doses of VSV (new panel - Figure S1F). Results section was amended accordingly (Page 5, lines 130-133).

6. I appreciate why the authors used suboptimal virus doses (1e6 PFU) in the CT26-LacZ model given that it is relatively easy to achieve cures in this tumour model with the standard dose (5e8 PFU). However, it has been shown that there is a threshold dose of virus needed to overcome the barriers to systemic delivery of OV. If the second dose is increased to 5e8 PFU, would this overcome the antiviral activity of the neutrophils present in the tumour microenvironment and lead to tumour infection with the second dose, and still be enhanced by monocytes?

We appreciate the scientific importance of the experiments suggested by the Reviewer. However, we have designed the experiments to utilize an OV dose that is focused on reducing the risk of system inflammatory storm responses. Although it would be interesting to determine if a threshold effect is present in this system, the overall strategy of using a dose that maximizes safety while still impacting tumour immunity is the central goal of the current studies. Perhaps more central to our choice of dosing is the fact that at higher dose of OV, the virus-sensitive CT26 model supports robust viral infection, a response that unfortunately is far different from many of the clinically responses where limited viral infection of the tumour is observed. Importantly we have also confirmed the observed post-infection conditioning in more resistant M3-9M model, highlighting the biological impact of these findings and demonstrating that the observed effects are not due to the highly infectable CT26 model.

7. How does the 2nd dose of VSV lower MIP-2 levels? What cells are making MIP-2? I would have thought macrophages/monocytes, but depletion of these cells with CLL increases MIP-2 levels, suggesting it is not the case. Also, can you block MIP-2 with neutralizing antibodies to show that the chemokine is critical for the enhancement effect of the 2nd dose?

In response to infection, MIP-2 can be produced by a variety of cells, including macrophages, monocytes, epithelial cells, and hepatocytes (PMID: 28533661). Unfortunately, we do not have data to identify the cellular source of MIP-2 in VSV-treated mice. We also cannot rule out the possibility that MIP-2 is not the

only (may be not the major?) factor that mediates the enhancement effect of the 2nd dose. Among 25 cytokines tested, we also found decrease in level of G-CSF in tumor (but not in blood) after 2nd dose administration (see below Figure 1 for revision only). Apart from the role of soluble factors, we show in the paper that monocytes directly interact with neutrophils changing their behavior making it unlikely that inhibition of single factors will re-capitulate the full phenotype. The differences in MIP-2 levels coupled with the changes in neutrophil number and behavior after 2nd treatment are likely both important and point towards the important role of neutrophils in 2nd dose effect, but do not yet identify a singular molecular mechanism. The latter remains the work in progress.

Figure 1 (for revision only). G-CSF response in tumor interstitial fluid (A) and blood (B) to a single or repeated VSV treatment at 56 h following single VSV treatment or 8 h following a second i.v. injection of VSV (10^6 PFU) \pm CLL 24 h post first virus administration; mean \pm SEM; n=4 (one-way ANOVA followed by Tukey's multiple comparisons test).

8. The authors demonstrate that neutrophils limit virus replication in both the 1st and 2nd dose, suggesting that these cells have antiviral activity. How do neutrophils limit VSV replication in this model, especially given that CT26-LacZ cells are interferon non-responsive? It does not seem to be via phagocytosis of virus particles as less virus associates with neutrophils in the 2nd dose compared to the 1st dose. Do they induce an IFN-independent antiviral state in tumour cells? Are neutrophil extracellular traps sequestering virus? Does the interaction between monocytes and neutrophils change gene expression of the neutrophils? Additional experiments aimed at answering this question would provide mechanistic insights on the role of neutrophils during OV therapy.

Neutrophils are extensively armed to fight invading pathogens including viruses [PMID: 23178588; PMID: 29327081]. Direct antiviral effect of neutrophils is thought to be associated with phagocytosis, NETs formation, ROS production, proteolytic enzymes release. In particular, human neutrophil peptide 1 inactivates VSV and other viruses in vitro (PMID: 3023659). Of note, our results are consistent with the previous study demonstrating the enhanced VSV infection in the same tumor model (interferon non-responsive CT26-LacZ) following neutrophil depletion (PMID: 17579581). Although the current paper

does not aim to identify the exact mechanism responsible for limiting virus replication by neutrophils, it can be suggested that this effect is likely associated with direct killing of infected cells and inactivation of the released virions by neutrophils rather than inducing antiviral state in the tumor cells. We agree with the Reviewer, that the potential mechanisms mediating neutrophils antiviral effect should be discussed. We have amended the Discussion section accordingly (page 20 , lines 459-463).

9. It has been shown that neutrophils mediate vascular shutdown following VSV infection in the CT26-LacZ model (PMID: 17579581). Can you rule out that this is not the reason there are fewer neutrophils in the tumour bed? If not, this should be elaborated in the discussion

This is a very good point. We have studied a possible role of the vascular shutdown in both 1 and 2 doses schedules. Although we routinely use IVM to study coagulation (PMID: 28073784; PMID: 27525062) we never observed a decrease in vessel densities or signs of clot formation in tumors during 24-72h after VSV injection. We also want to point the Reviewers attention to our FC and IVM data showing that a decrease in number of neutrophils after 2nd VSV treatment (Figure 4D) is not accompanied by the decrease in number of CD8 cells (Figure 5E, G) or monocytes (new panel Figure S2F). It is reasonable to expect that in case of vascular shutdown these leukocyte subsets should also decline. We too were surprised we did not see the vascular shutdown described by Breitbach et al. in the same tumor model (PMID: 17579581) and believe it is most likely due to differences in the injected dose: vascular collapse was documented in animals treated with 5e8 pfu VSV, while in the current study we used a 500-fold lower dose. We included these considerations in the revised version of the manuscript (page 20 , lines 456-459).

10. The finding that CD8 depletion improves VSV replication is quite interesting. Can you assess VSV binding to CD8 T cells to determine if CD8 T cells do indeed exert antiviral activity against VSV as opposed to simply binding and sequestering virus particles away from the tumour. The difference is there even 24h after the first dose, and at this point the T cells are just becoming activated and finished blasting, so I am not too sure how much effector functions they have this early on after infection.

We thank the Reviewer for this interesting comment. We have analyzed VSV capturing by CD8 T cells in blood and spleen (updated panels S2A, S2B) and found that less than 1% of CD8 cells are capable of binding the virus.

Therefore, it is unlikely that CD8 cells sequester virus particles away from the tumour. We agree with the Reviewer, that early effect of CD8 depletion argues against their specific antiviral activity. Notably, the enhanced replication of Newcastle Disease Virus in B16 tumors of CD8-depleted mice has been shown previously, although the mechanism remains elusive. To further investigate the role of CD8 cells in tumor infection we analyzed cytokines in tumor interstitial fluid with and without CD8 depletion (new Figure S5E). The results demonstrate a decrease in the levels of proinflammatory cytokines and chemokines (IFN- γ , IL-6, G-CSF, KC) in CD8-treated mice that could potentially explain the enhanced viral spread. Although CT26-LacZ cells are non-responsive to IFNs, non-cancer cells of tumor microenvironment that are permissive for VSV replication (endothelium (PMID: 21364541); CAFs (PMID: 25894825)) may render more susceptible to the virus in IFN-depleted milieu. Of note, *in vitro* IL6 is shown to repress the replication of classical swine fever virus (PMID: 19923180) and hepatitis B virus (PMID: 19374779) in interferon-independent manner. Moreover, decrease in G-CSF and KC could result in less efficient recruitment of neutrophils unleashing the infection in the tumor. Results section was amended to provide potential explanation for more robust infection in animals pretreated with anti-CD8 antibodies (page 16, lines 357-359).

11. Please verify the p-value in Figure S2B? The CD11b+ Ly6G hi group seems like it should be significant based on the 2 previous groups.

We thank the Reviewer for careful reading. The p-values on Fig S2B have been corrected.

12. I don't see the value in Figure 1D as it is, it simply shows that VSV replicated in the tumour which you have already established. A more appropriate figure would be to compare 1 dose of VSV-GFP, to 2 doses with 1st dose being VSV-NR and 2nd dose VSV-GFP

The difference between 1 dose of GFP and 2 doses with 1st dose being VSV-NR and 2nd dose VSV-GFP is shown on Figure 1H. We used different VSV constructs to demonstrate that the observed effects are consistent between viruses expressing different transgenes. For this reason, we would like to keep Figure 1D as it is.

13. The y-axis in Fig 4D seems cut-off by the y-axis title

Thanks for the comment. The problem is now fixed

14. Normally in an uninfected mouse there are more CD4 T cells than CD8 T cells (the ratio is somewhere around 3:2 CD4:CD8), but in Figure S2C and S2E the percentage of CD4 T cells is extremely low (less than 1% CD4 T cells, but 10-15% CD8 T cells). This experiment would have to be repeated as I doubt this is truly the case.

New experiments were performed to analyze CD4+ cell numbers in blood and tumors of untreated and VSV-treated mice. Updated panels S2C and S2E show CD4 numbers more closely parallel what is expected.

Referee #2

1. The figures in this manuscript are presented clearly and are easy to follow, but it would be beneficial to the reader if the authors could include the gating strategy for the FACS data, and clarify what they have termed neutrophils in figure 4.

Thanks for the comment. We have added new figure S7 to show gating strategy for FC, in particular to clarify what do we call neutrophils in figure 4.

2. Furthermore in Figure 4 A MIP-2 should be presented as a concentration not fold increase.

The corresponding changes have been made.

3. Could the authors also explain the discrepancy in the survival experiment in 4L as the two doses do not appear to match previously presented survival curves in Figure 1B, 3F and 5B (it would be also helpful if axis were kept consistent throughout the manuscript).

Following the Reviewer's comment, we repeated the experiment shown in 4L and the results were fully recapitulated including the lack of long-term survivals in animals treating with 2 doses (see below Figure 2 for revision only). We think that the discrepant results on the antitumor efficiency of VSV between experiments is due to the low dose ($1e6$) that is marginal for providing survival benefits and can be significantly influenced by inevitable variability between different experiments. The axis for survival graphs are now consistent throughout the manuscript.

Figure 2 (for revision only). Tumor measurements (A) and Kaplan-Meier survival plots (B) for CT26^{LacZ}-bearing mice treated with two doses of VSV-FLUC (10^6 PFU, 48 h between i.v. injections) \pm anti-Ly6G 24 h after the first dose of VSV; mean \pm SEM (n=8).

4. The authors generally describe their findings in a concise and accurate manner however there is one incidence where the sentence is a bit misleading. Line 202 'neutrophils are absent from the tumour 24hrs following the second dose' they are not absent but just return to untreated controls (also the axis related to this figure has been cut off).

We thank the reviewer for this comment. The sentence has been rephrased (page 12 , lines 268-269).

5. In the discussion the authors fail to discuss that the model that they have used is immunologically 'hot' and therefore priming T cells response is possible following OVT, they focus instead more on the onolytic nature despite their results showing that even if you do enhance replication that does not lead to better outcomes (figure 3F vs Figure 4J). Do the authors have anymore information on how this multiple dose regimen performs in resistant or 'cold' tumours?

We agree with the Reviewer, that both our models (CT26 and M-3-9M) are immunologically "hot" and this is one of the potential limitations of the study. As such, it is discussed in the revised version of the manuscript (page 21 , lines 484-488). We plan of expanding our research into less immunogenic tumours in the future but we feel this work is beyond the scope of the basic principles and mechanisms being discussed

Referee #3

1. One suggestion for improvement is that the accompanying videos are difficult to follow and the authors should make an attempt to better annotate them. It is also suggested that they should slow the speed of video replay to allow better tracking of individual cells and viruses. That said, the intravital microscopy videos are an appealing aspect of the paper.

We thank the Reviewer for this comment. The videos have been updated and we hope that now they are more informative and easier to watch.

Referee #4

1. The authors conducted a TCID₅₀ in figure 1 but did not follow up on the virus fitness through the various experiments that manipulated the immune component (i.e., anti-CCR2, or anti-Ly6G). The addition of this would significantly strengthen the author's claims

Additional experiments were performed to evaluate virus fitness in the tumors of Ly6G-depleted mice. The results are shown in Figure S4D. Although the average virus titers were 6.7-fold higher in Ly6g-depleted mice as compared to control, the difference was not statistically significant due to high variability between tumors (despite having 9 mice in each group).

2. Does a 2nd dose increase monocyte function or accumulation in comparison to a single dosage? It would be important to see if monocyte accumulation is greater or cytokine production is higher in 2nd dose monocytes

We agree with the Reviewer that it is of importance to understand if 2nd dose further increases monocyte numbers and/or activates the cells. To address the question, we analyzed monocyte numbers in blood and tumors 72h after 1st dose injection vs 72h/24h after 1st/2nd treatment using FC (new Figures S2F). The results suggest that the 2nd dose fails to further activate monocyte recruitment. The results section was amended accordingly (page 10 , lines 20-22).

3. For flow cytometry analysis a representative gating strategy is essential for determining whether the immune cells are properly identified. This is particularly important for markers like CD169+ Marginal metallophilic macrophages in the spleen which are notorious for not being captured by flow cytometry

New Figure S7 now shows gating strategy for immune cells in tumor, blood, and spleen. Indeed, identification of CD169+ metallophilic macrophages in the spleen is challenging, however, tumor CD169+ macrophages are easily detected.

4. Do the authors observe a difference in Red pulp macrophages and marginal metallophilic macrophages?

Our data show an increase in number of both F4/80+ cells and CD169+ cells in the spleen 48h after VSV injection (Figure S2D). Interestingly, while the ability of F4/80+ cells to capture 1st dose or 2nd dose remains unchanged, CD169+ cells bind 1st dose more efficiently than 2nd dose (Figure S2B).

5. Please give a rationale for the depletion of CD8+ T cells prior to OV treatment. Also, what is the frequency of CD8+ T cells in nontreated tumors? Does a single dose also have a benefit?

To better understand the mechanism mediating 2nd dose effect we sought to investigate if 2nd dose enhancement of infection provides survival benefits per se, in the absence of CD8 T cells. Unexpectedly we found a dramatic increase in tumor infection in VSV-treated mice upon CD8 depletion. However, CD8 depletion completely abrogated the therapeutic efficacy of the therapy. In agreement with previous reports, these results indicate that the ability to elicit CD8 T cells response is more important for OVT than direct oncolysis. Next, we studied if 2nd dose influences the recruitment of CD8 T cells to the tumor. The frequencies of CD8+ T cells in untreated mice and in differentially treated groups are shown in Figure 5E. Although a single dose provides some survival benefit (Figure 1B), our results indicate that significant increase in number of CD8 T cells (and further increase in survival) is achieved only in animals receiving 2 doses of VSV. We made changes in the Results section to better explain the rationale of CD8 depletion experiments (page 16, lines 345-347).

6. Do the CD8+ T cells have different functional aspects following a single or 2 dose regimen of OV?

We agree with the Reviewer that functional activity of CD8 T cells is important for the success of the therapy. To address this question, we used cellular behavior as a surrogate measure of CD8+ T cells function at 72h after 1st dose injection vs 72h/24h after 1st/2nd treatment. Our IVM results (new figure S5F) indicate that although the total number of intratumoral CD8 cells is increased in mice receiving 2 doses, there is no difference in cellular behavior between groups. With that said, we understand that further studies are needed to evaluate other aspects of CD8 activity in single vs 2 doses OVT regimens. Results section was amended to describe the data (page 16-17, lines 362-364).

7. Would recommend changing the title to Repeated dosing improves...

We thank the Reviewer for the suggestion. The title has been changed.

REVIEWERS' COMMENTS:

Reviewer #1 (Remarks to the Author):

I thank the authors for careful consideration of my comments and their rebuttals, and commend them on their improved manuscript.

Most of my concerns were addressed in the rebuttal and the new data strengthens the manuscript's conclusions.

I have no further comments or questions for the authors.

Reviewer #2 (Remarks to the Author):

I am happy with the modifications to text and figures highlighted in my original review of this manuscript, the authors have covered all the points and have addressed them to an acceptable level.

Reviewer #4 (Remarks to the Author):

The authors have addressed my concerns.